# THREADING KEYFRAME WITH NARRATIVES: MLLMS AS STRONG LONG VIDEO COMPREHENDERS

**Bo Fang[1], Yuxin Song[2], Haoyuan Sun[2,3], Qiangqiang Wu[1], Wenhao Wu[4], Antoni B. Chan[1]***
[1]City University of Hong Kong, [2]Baidu Inc, [3]Tsinghua University, [4]The University of Sydney
bofang6-c@my.cityu.edu.hk, abchan@cityu.edu.hk

## ABSTRACT

Employing Multimodal Large Language Models (MLLMs) for long video under-standing remains a challenging problem due to the dilemma between the substantial number of video frames (i.e., visual tokens) versus the limited context length of lan-guage models. Traditional uniform sampling often leads to selection of irrelevant content, while post-training MLLMs on thousands of frames imposes a substantial computational burden. In this paper, we propose *Narrating KeyFrames Capturing* (**Nar-KFC**), a plug-and-play module to facilitate effective and efficient long video understanding. Nar-KFC generally involves two collaborative steps. First, we formulate the *keyframe* selection process as an integer quadratic programming problem, jointly optimizing query-relevance and frame-diversity. To avoid its computational complexity, a customized greedy search strategy is designed as an efficient alternative. Second, to mitigate the temporal discontinuity caused by sparse keyframe sampling, we further introduce interleaved textual *narratives* generated from non-keyframes using off-the-shelf captioners. These narratives are inserted between keyframes based on their true temporal order, forming a coherent and compact representation. Nar-KFC thus serves as a temporal- and content-aware compression strategy that complements visual and textual modalities. Experimental results on multiple long-video benchmarks demonstrate that Nar-KFC significantly improves the performance of popular MLLMs. Codes are publicly available at https://github.com/bofang98/Nar-KFC.

## 1 INTRODUCTION

Building upon the success of revolutionary Large Language Model (LLMs) (Touvron et al., 2023; Team et al., 2024), recent advances in Multimodal Large Language Models (Liu et al., 2023; Li et al., 2024b; Wang et al., 2024b; Chen et al., 2024c; Tong et al., 2024; Lin et al., 2024b) have significantly improved open-world visual understanding. Moving beyond static images, a natural extension of MLLMs is their application to video understanding. Existing studies have validated their effectiveness in comprehending *short* videos ($\sim$10 s) (Yang et al., 2022; Kim et al., 2024; Yao et al., 2024a). However, when scaling MLLMs to long videos (Fu et al., 2025; Wu et al., 2024b; Chandrasegaran et al., 2024; Zhou et al., 2025) (e.g., hours), several critical challenges emerge.

The primary challenge stems from the inherent context limitation of MLLMs, which cannot accom-modate the vast volume of visual tokens generated from the whole video. A prominent solution is to extend the context window of language models and fine-tune them on carefully collected long videos. Current video-oriented LLMs, known as VideoLLMs (Lin et al., 2024a; Jin et al., 2024; Song et al., 2024; Xu et al., 2024a; Chen et al., 2024b; Zohar et al., 2024; Shu et al., 2025; Cheng et al., 2025a; Wang et al., 2025a), typically undergo post-training on existing LLMs/MLLMs through: 1) employing a relatively large stride uniform sampling scheme, and 2) incorporating token-level merging or compression techniques to enable broader temporal coverage. However, uniform sampling often fails to preserve key moments relevant to specific instructions, while feeding an excessive number of frames as input introduces redundancy, leading to substantial computational overhead. An alternative solution follows a training-free paradigm (Zhang et al., 2024a; Kahatapitiya et al., 2024; Wang et al., 2024d; 2025b; Park et al., 2024; Ma et al., 2025), where raw videos are first converted

---

*Corresponding author.

into sequential captions, which are subsequently processed using the long-range reasoning abilities of LLMs (Achiam et al., 2023). Compared to direct video frame encoding, textual captions inherently require far fewer tokens, allowing efficient inference in a single forward pass. Nonetheless, the translation from video frame to caption inevitably results in critical information loss (e.g., important visual features), potentially leading to hallucinated answers caused by the LLM bias.

Regarding the aforementioned paradigms, e.g., training a VideoLLM or reasoning with LLMs on textual captions, are current MLLMs fully equipped to comprehend long videos despite their limited context length? Instead of relying on uniform sampling, recent studies have focused on learning to select query-relevant keyframes (Yu et al., 2023; Hu et al., 2025; Yao et al., 2025) to facilitate inference with MLLMs. Due to the temporal redundancy among adjacent frames, trival similarity-based keyframe selection tends to retrieve frames located within narrow time windows, thereby compromising accuracy. To this end, adaptive keyframe sampling (Tang et al., 2025), inverse transform sampling (Liu et al., 2025b), DPP sampling (Sun et al., 2025) have been proposed to promote content diversity to mitigate the concentration of keyframes. Despite a decent boost over existing MLLMs, these methods largely depend on handcrafted or heuristic strategies with limited theoretical formulations, and empirically, the retrieved frames can be temporally distant, especially in long videos. Consequently, the keyframe selection process can introduce temporal discontinuities into the input provided to the MLLM, ultimately hindering its holistic understanding of video content.

In this paper, we propose *Nar-KFC* (**Nar**rating **K**ey**F**rames **C**apturing), a training-free framework for long video understanding with MLLMs. Unlike previous approaches, Nar-KFC jointly considers *query-relevance*, *frame-diversity* and *temporal-continuity* through two collaborative stages. The first stage **KFC** selects keyframes by considering both query relevance and frame diversity, so as to resolve the issues of critical information loss from uniform sampling and the too-narrow focus using just query-relevance. We consider keyframe selection as a graph problem, where each node is a frame and the edge weight (score) between nodes combines query-relevant similarities and frame-to-frame dissimilarities (frame-diversity). The optimal keyframes are obtained by finding the subgraph with largest total edge weight, which can be formulated as an integer quadratic programming (IQP) problem. However, since IQP is NP-hard with exponential complexity, finding exact solutions is infeasible in practice. To overcome this, we introduce a robust and efficient greedy search (GS) strategy, which, with proper preprocessing of the score matrix, achieves near-optimal performance with significantly reduced computational complexity.

The second stage **Nar-KFC** addresses the problem of temporal discontinuities caused when selecting keyframes at uneven timestamps. Specifically, Nar-KFC works by threading keyframes (visual tokens) with *non-keyframe narratives* (text tokens), generated by captioning the intermediate, unselected frames in between, aiming to reconstruct the video as a continuous and coherent sequence in both textual and visual modalities. A narrative interval is further applied to control the total number of captions and to reduce the similarity between neighboring descriptions. Leveraging only a lightweight 2B captioning model, e.g., Qwen2-VL-2B (Wang et al., 2024b), Nar-KFC demonstrates significant improvements over existing MLLMs. In summary, the contributions of this paper are three-fold:

- Jointly considering query-relevance and frame-diversity, we formulate the keyframe capturing process (KFC) in long videos as a subgraph selection problem, implemented as an integer quadratic programming problem. We introduce a customized greedy search algorithm to solve this problem with significantly reduced and practical time complexity.
- We propose Nar-KFC, which threads the optimized keyframes with non-keyframe narratives. By interleaving the two modalities in a temporally continuous manner, Nar-KFC constructs coherent and compact video representations, enabling a broader video coverage under the constraint of frame length limitations in current MLLMs.
- Our KFC and Nar-KFC are generally compatible with many MLLMs, achieving consistent improvements across four mainstream MLLMs on multiple long-video benchmarks.

## 2  RELATED WORK

Transformer-based LLMs have revolutionized the field of natural language processing (Brown et al., 2020; OpenAI, 2023; Grattafiori et al., 2024; Achiam et al., 2023). By incorporating multimodal inputs such as images and videos (Li et al., 2024b; Zhu et al., 2023) with a vision encoder, e.g., ViT (Dosovitskiy et al., 2020), researchers further extend powerful LLMs to multimodal large

language models (MLLMs) for open-world visual understanding (Alayrac et al., 2022; Li et al., 2023a; Liu et al., 2023). Despite similar advancements of MLLMs on various video understanding tasks including video captioning (Chen et al., 2024a; Yang et al., 2023; Wu et al., 2024a), video question answering (Maaz et al., 2023; Li et al., 2023b; Min et al., 2024), and temporal reasoning (Qian et al., 2024), significant challenges emerge when scaling to long videos due to the substantial amount of video frames not fitting in the limited context length of LLMs (Wu et al., 2024b).

Recent studies have explored methods to extend the context length of LLMs (Wan et al., 2024; Xiong et al., 2024), or introduced various token-level merging and compression techniques (Song et al., 2024; Shen et al., 2024; Li et al., 2024d; Wang et al., 2024c; Shu et al., 2025) to accommodate more frames as input. However, these approaches typically require additional fine-tuning of existing language models, which increases computational complexity and introduces the risk of hallucinations (Liu et al., 2024c). Given that textual tokens are significantly fewer than visual frames, another line of research first converts all video frames into textual descriptions, which are then used for long video inference, either by summarizing them (Zhang et al., 2024a; Park et al., 2024) or identifying central frames based on textual similarity via agents (Wang et al., 2024d; 2025b; Ma et al., 2025; Ye et al., 2025; Liu et al., 2025a). Nonetheless, the converting process inevitably leads to critical information loss, thereby compromising performance. Other studies, while maintaining the number of input frames, adopt alternative sampling strategies instead of default uniform sampling to obtain higher-quality frames for input. In general, query relevance is the primary criterion for selecting frames that are semantically closest to the query (Yu et al., 2023; Lin et al., 2024b; Wang et al., 2024d;a; Suo et al., 2025). Methods such as AKS (Tang et al., 2025), BOLT (Liu et al., 2025b), Frame-Voyager (Yu et al., 2025) further propose adaptive sampling, inverse transform sampling, and optimal frame combination sampling to identify keyframes that are both query-relevant and temporally distinctive. Nevertheless, the methods often rely on manually designed heuristics without principled theoretical guidance, and the selected keyframes are often undistributed and distant over long intervals, especially in hours-long videos (e.g., 3600 frames per hour at 1 fps). This temporal sparsity weakens the relationships between frames and can cause confusion in MLLM inference.

In contrast to previous works, we formulate long video keyframe selection as a graph-based optimization problem with a clearly defined objective, and further leverage the efficiency of textual descriptions. Our approach jointly considers query relevance, content diversity, and temporal continuity, aiming to construct optimal combinations of keyframes with interleaved narratives, under the constraints of MLLM context length.

## 3 METHOD

### 3.1 KFC: KEYFRAME CAPTURING

Uniform sampling is commonly used in *short* video understanding for consistent temporal structure. However, for *long* videos, it often misses important information with limited input. While recent works emphasize selecting query-relevant frames for long video QA, they tend to overlook the problem of narrow focus due to the high similarity between adjacent frames. To address this, we first propose a keyframe capturing method that simultaneously considers query-relevance and frame-diversity, modeling the selection process as subgraph selection problem.

**Preliminaries.** General video understanding tasks, e.g., video summarization and grounding (Liu et al., 2024b; Xiao et al., 2024) and long-video QA, can be similarly formulated as $(V, q) \rightarrow$ Answer, where $V = \{f_i\}_{i=1}^{N}$ represents a video with $N$ frames, $f_i$ is the $i$-th frame, and $q$ is the query. Considering an MLLM model as a neural function $\mathcal{M}(\cdot)$ with its limited contextual perceiving length, the normal video QA process reasoned by an MLLM model can be formulated as $\mathcal{M}(\{f_i\}_{i=1}^{K}, q) \rightarrow$ Answer, $1 \leq K \ll N$, meaning that only $K$ frames are captured for representing video $V$. We next consider two criteria for selecting the $K$ frames, query-relevance and frame-diversity.

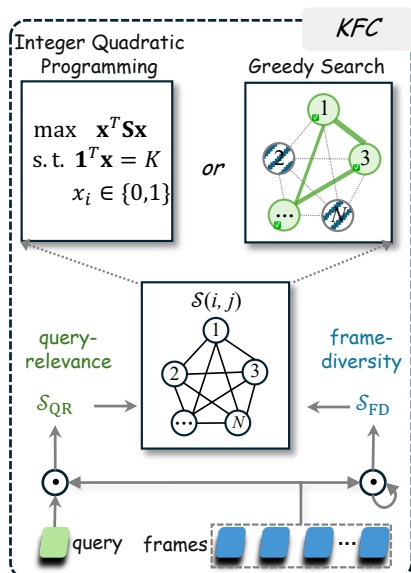

Figure 1: Illustration of keyframe capturing (KFC). $\mathcal{S}_{\mathrm{QR}}$ and $\mathcal{S}_{\mathrm{FD}}$ scores are computed via inner dot production.

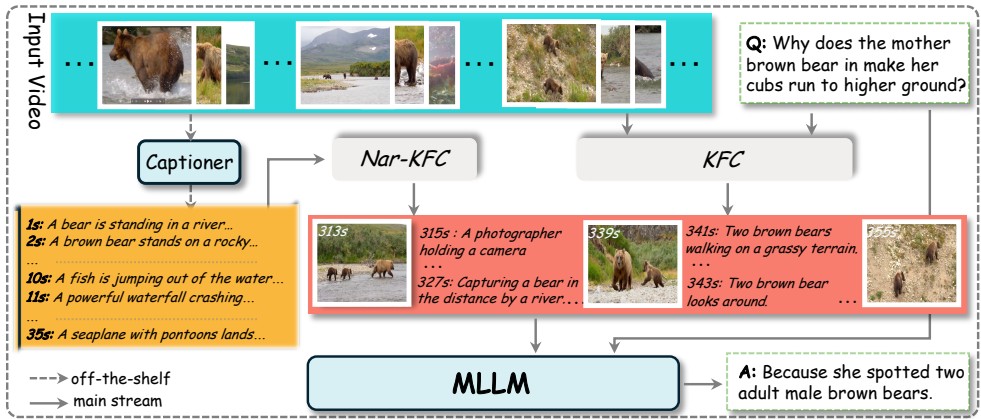

Figure 2: Illustration of Nar-KFC. We represent long videos by threading KFC-optimized keyframes with temporally interleaved narratives, where the narratives are generated frame-wise by an off-the-shelf captioner. Nar-KFC constructs a continuous representation to facilitate MLLM inference.

**Query-relevance.** Since different questions can be asked on the single video, it is crucial to identify frames that correspond to a specific query first. Here, a standard two-stream vision-language model (VLM), e.g., CLIP (Radford et al., 2021), is used to extract embeddings $\{\mathbf{f}_i\}_{i=1}^N$ and $\mathbf{q}$ for the frames and the query, respectively. After standard normalization of all embeddings, the query-relevance score $\mathcal{S}_{\text{QR}}$ is computed as the cosine similarity between the two, $\mathcal{S}_{\text{QR}}(i) = \text{sim}(\mathbf{f}_i, \mathbf{q})$.

**Frame-diversity.** To avoid retrieving query-relevant only frames that are narrowly located in a small time range, we explicitly encourage diversified content when choosing the $K$ frames. In particular, we use the *inverse* of cosine similarity between every pair of frame embeddings (normalized) to represent the diversity score. The function $\exp(\cdot)$ is applied to constrain the score between 0 and 1, formulated as $\mathcal{S}_{\text{FD}}(i, j) = \exp(-\text{sim}(\mathbf{f}_i, \mathbf{f}_j))$.

**Objective.** The final score combines $\mathcal{S}_{\text{QR}}$ and $\mathcal{S}_{\text{FD}}$ to jointly identify keyframes that are both query-relevant and diversified for KFC,

$$\mathcal{S}(i, j) = \mathcal{S}_{\text{QR}}(i) + \mathcal{S}_{\text{FD}}(i, j) = \text{sim}(\mathbf{f}_i, \mathbf{q}) + \exp(-\text{sim}(\mathbf{f}_i, \mathbf{f}_j)). \tag{1}$$

Next, as illustrated in Fig.1, we construct a graph where each node is a frame, and the edge weight between node pair $(i, j)$ is $\mathcal{S}(i, j)$. The selection of $K$ keyframes can then be cast as a subgraph selection problem with the original objective as follows: *given $N$ nodes (frames), construct a subgraph by selecting $K$ nodes (keyframes) so as to maximize the total edge weight of the subgraph.* Mathematically, this objective can be expressed as the optimization problem:

$$\max_{Y \subset \{1, \cdots, N\}, |Y| = K} \sum_{(i,j) \in \mathcal{I}} \mathcal{S}(i, j), \tag{2}$$

where $Y = \{y_1, \cdots, y_K\}$ is the index set of the $K$ keyframes and $\mathcal{I}$ denotes all pairs $(i, j)$.

### 3.1.1 THEORETICAL OPTIMUM: INTEGER QUADRATIC PROGRAMMING

Our objective closely resembles the classic Knapsack problem (Salkin & De Kluyver, 1975), which can be commonly solved by dynamic programming or integer linear programming. The problem in (2) can be rewritten equivalently as an **integer quadratic programming (IQP)** problem,

$$\max_{\mathbf{x}} \ \mathbf{x}^T \mathbf{S} \mathbf{x} \quad \text{s.t.} \ \mathbf{1}^T \mathbf{x} = K, \ x_i \in \{0, 1\}, \tag{3}$$

where $x_i = 1$ indicates that the $i$-th frame is selected, $\mathbf{x} = [x_1, x_2, \cdots, x_N]^T$, and $\mathbf{S} \in \mathbb{R}^{N \times N}$ is the score matrix with $\mathbf{S}_{i,j} = \mathcal{S}(i, j)$ for $i < j$, and $\mathbf{S}_{i,j} = 0$ otherwise. Here, only the upper triangle of $\mathbf{S}$ is considered. A discussion of symmetrical $\mathbf{S}$ is detailed in Appendix §E.1. The search space is $C(N, K)$, and the time complexity of solving IQP is exponential regardless of whether the objective is convex or non-convex, making it impractical to get exact solutions in real cases. Modern optimization tools, e.g., CPLEX (Bliek1ú et al., 2014), typically address this by relaxing the binary

constraint and allowing $x_i \in [0, 1]$, converting the problem into a continuous optimization task. Solutions can then be obtained using methods like interior-point or Lagrange multiplier methods, with a complexity of $\mathcal{O}(N^3)$. Subsequently, the Branch & Bound algorithm (Morrison et al., 2016) is applied to prune the search space and retrieve optimal integer solutions of $x_i$, but the worst-case time complexity remains exponential.

### 3.1.2 PRACTICALLY FEASIBLE APPROACH: GREEDY SEARCH

Solving the IQP optimally is computationally intractable for large $N$, e.g., long videos with thousands of frames. To search keyframes within practical latency constraints, we propose an efficient **greedy search (GS)** strategy that yields robust and near-optimal effects to the IQP solution. We first pre-process the score matrix to reduce noise across adjacent columns/rows, and shrinks the problem size for greater computational efficiency. Specifically, we apply singular value decomposition (SVD) to the score matrix $\mathbf{S}$, retaining the top $r$ singular values to construct a low-rank approximation $\mathbf{S}_r \in \mathbb{R}^{N \times N}$. This matrix is then uniformly downsampled to $\mathbf{S}_{rd} \in \mathbb{R}^{\frac{N}{d} \times \frac{N}{d}}$ with a downsampling ratio $d$. The GS algorithm begins by selecting the most query-relevant frame as the starting point. It then iteratively adds the frame with the highest *cumulative* score relative to the already selected frames. In the final refinement step, the algorithm examines the $k$-nearest neighbors of each selected frame $y_i$, replacing $y_i$ with a neighboring frame if it yields a higher cumulative score based on $\mathcal{S}_r$. A summary of the algorithm is provided in Alg. 1, and its overall time complexity is $\mathcal{O}(NK)$.

---

**Algorithm 1:** Practically Feasible Approach with Greedy Search

**Input:** Query-relevant score $\mathcal{S}_{\text{QR}}$, score matrix $\mathbf{S}$, number of retained singular values $r$, downsample ratio $d$, number of frames $N$, neighbor window $k$.
**Output:** Indices of selected $K$ frames set $Y = \{y_1, y_2, \cdots, y_K\}$
1  $\mathbf{S}_r \leftarrow \text{LowRank}(\mathbf{S})$;   $\mathbf{S}_{rd} \leftarrow \text{Downsample}(\mathbf{S}_r, d)$; // Decompose and downsample $\mathbf{S}$
2  $y_1 = \arg\max_i \mathcal{S}_{\text{QR}}(i)$; $Y \leftarrow \{y_1\}$ // Initialize with most query-relevant frame
3  **for** $i \leftarrow 2$ **to** $K$ **do**
4      **for** $j \leftarrow 1$ **to** $N$ **do**
5          $y_i = \arg\max_j \sum_{y \in Y} S_{rd}(y, y_j)$ // Select frame with highest sum
6          $Y \leftarrow Y \cup y_i$
7  **for** $i \leftarrow 1$ **to** $K$ **do**
8      $y_i = \text{Refine}(y_i, k | \mathbf{S}_r)$;   // Refine selection within $k$-nearest neighbors
9  **return** $Y = \text{sorted}\{y_1, y_2, \cdots, y_K\}$;

---

### 3.2 NAR-KFC: THREADING KEYFRAME WITH NARRATIVES

Keyframes captured by KFC significantly enhance the performance of MLLMs compared to the default uniform inference mechanism. However, it overlooks the *temporal-continuity* in frame sequences. Due to the severely uneven distribution of selected frames, temporal relationships become weak, often leading to confusion during inference.

To this end, we propose **Nar-KFC**, which threads keyframes with text narratives to construct a continuous and coherent input in an interleaved form. Specifically, we first use a lightweight off-the-shelf captioner, e.g., Qwen2-VL-2B, to generate captions $\{c_i\}_{i=1}^N$ for non-keyframes using a simple prompt as "`<USER> Describe this video frame in no more than 15 words.`" Given the unevenly distributed keyframes $\{f_{y_i}\}_{i=1}^K$ from KFC, we insert *captions from non-keyframes* between the keyframes, arranging them according to their true temporal order. Each $y_i$ denotes the timestamp, and a uniform interval $\triangle$ is set between captions to control the total number of inserted narratives. The overall long video inference to a MLLM model $\mathcal{M}$ is formulated as:

$$\mathcal{M}\big(\{f_{y_1}, c_{y_1+\triangle}, \cdots, c_{y_2-\triangle}, f_{y_2}, c_{y_2+\triangle}, \cdots, c_{y_K-\triangle}, f_{y_K}\}, q\big) \rightarrow \text{Answer.} \quad (4)$$

**Viability** of Nar-KFC. MLLMs are typically trained via instruction tuning on both visual and textual modalities, making them well-suited to process our interleaved inputs of keyframes and narratives. **Rationality** of Nar-KFC. The approach provides a temporally continous input that helps MLLMs "narrate" the story between keyframes. From another perspective, Nar-KFC can be seen as a form of compression, retaining only the most informative keyframes, while representing less critical

segments with brief textual descriptions. This complementary two-stream mechanism is analogous to method like Two-Stream (Simonyan & Zisserman, 2014), which combines RGB frames with optical flow. Also, it shares conceptual similarities with SlowFast (Feichtenhofer et al., 2019) and SlowFast-LLaVA (Xu et al., 2024b), where the caption stream serves as *a fast branch* traversing a broader temporal range (as in the low frame rate of the slow branch in SlowFast). These mechanisms together help explain the effectiveness of Nar-KFC in (long) video understanding.

## 4 EXPERIMENTS

### 4.1 EXPERIMENT SETTINGS

**Evaluation Benchmarks.** We evaluate our methods on several widely-used long-video question-answering benchmarks:: 1) *Video-MME* (Fu et al., 2025), consisting of 2,700 human-annotated QA pairs, with an average video duration of 17 min; 2) *LongVideoBench* (Wu et al., 2024b) validation set (denoted as *LVB*), which contains 1,337 QA pairs with average duration of 12 min; 3) *MLVU* (Zhou et al., 2025), where we use the multiple-choice task (M-avg), comprising 2,593 questions across 9 categories, with an average duration of 12 min. Besides, open-ended generation performance on MMBench-Video (Fang et al., 2024) and MLVU-OpenEnded (Zhou et al., 2025) (G-avg) are evaluated, to verify the fine-grained capabilities of our methods. We provide more results of on relatively short EgoSchema (3 min) (Mangalam et al., 2023) and NExTQA (44 sec) (Xiao et al., 2021) benchmarks in Appendix §D.3.

**Evaluation Models.** We consider multiple advanced MLLMs, including InternVL2 (Chen et al., 2024c), Qwen2.5-VL (Bai et al., 2025b), LLaVA-OneVision (Li et al., 2024b), LLaVA-Video (Zhang et al., 2024d), InternVL3 (Zhu et al., 2025), and Qwen3-VL (Bai et al., 2025a), to verify the effectiveness of our methods. We re-implement baseline results (uniform sampling) of these MLLMs using VLMEvalKit (Duan et al., 2024), which may yield slight differences compared to other public toolkits, e.g., LMMs-Eval (Li et al., 2024a).

**Implementation Details.** We use CLIP-ViT-L-336px (Radford et al., 2021) to extract query and video frame embeddings. Candidate frames are sampled from raw videos at 1 fps. For solving the IQP, we limit the maximum search nodes to 40k in CPLEX (Bliek1ú et al., 2014). In our customized greedy search algorithm, we empirically retain the top $\frac{N}{4}$ singular values to form the low-rank approximation of the score matrix $\mathbf{S}$ and further downsample it to a fixed resolution of $128 \times 128$ following previous work (Yu et al., 2025; Sun et al., 2025). The refinement window size $k$ is set to 2 (see ablations of hyperparameters in Appendix §E.3). Unless otherwise stated, all ablations are conducted using the InternVL2 model on Video-MME. Experiments are run on 8 A100 GPUs.

### 4.2 BENCHMARK RESULTS

**Comparisons with State-of-the-Arts.** We conduct comprehensive comparisons between our approach and several recent MLLMs and VideoLLMs in Tab. 1. Earlier works, e.g., Video-LLaVA (Lin et al., 2024a), Chat-UniVi-V1.5 (Jin et al., 2024), VideoLLaMA2 (Cheng et al., 2024), *etc*, are fully included in Appendix §D.1. Our methods, KFC and Nar-KFC, deliver consistent and significant gain over five baselines across three long-video benchmarks. On *Video-MME* (no sub.), Nar-KFC outperformsfive MLLM baselines by 4.38% in average. Using the strongest baseline, i.e., InternVL3, Nar-KFC achieves state-of-the-art performance (63.8%), surpassing previous VideoLLMs - even those using larger LLMs (e.g., VILA-34B, 58.3%) or more frames (e.g., Video-XL$^{256\mathrm{frm}}$, 55.5%). Incorporating larger numbers of frames may introduce noise and irrelevant information, which can be well addressed by our keyframe capturing and narrating strategies. On *LVB*, our method also achieves notable performance improvements, e.g., 52.3% *vs.* 53.9% with InternVL2 and 52.7% *vs.* 55.3% with Qwen2.5-VL, although the overall gain is partly offset by videos shorter than 1 min, demonstrating clear advantages in long video understanding. On *MLVU*, our KFC-only strategy (without narrations) yields an average improvement of over 6% across five MLLMs. The use of query-relevant and diverse keyframes significantly boosts performance on Needle-in-a-haystack (Zhang et al., 2024b) and counting questions. Furthermore, appending narratives provides additional and robust gains by preserving temporal continuity. Detailed analysis are further presented in Appendix §D.4.

**Comparisons with varying number of keyframes.** In Fig. 3, we compare KFC and Nar-KFC against uniform sampling with varying frames across three benchmarks and three models. Due to Qwen2.5-VL's dynamic resolution mechanism (Dehghani et al., 2023), increasing keyframes often leads to memory overflow, so its results are omitted. Notably, Nar-KFC shows substantial gains when the number of keyframes is limited (e.g., 4 or 8), due to its ability to provide broad video

Table 1: Comparisons with previous VideoLLMs on: Video-MME, LVB, and MLVU. All methods are evaluated using **8** frames. For Video-MME, we report performance with two standard settings: without subtitles (no sub.) and with subtitles (sub.). LVB denotes the LongVideoBench set. Methods that use significantly more frames and larger-sized LLM are marked in gray. The reported results are accuracy percentage.

| Model | Size | Video-MME(no sub. / sub.) | | | | LVB | MLVU |
|---|---|---|---|---|---|---|---|
| | | Short | Medium | Long | Overall$_{\sim17m}$ | $\sim12m$ | $\sim12m$ |
| VILA (Lin et al., 2024b) | 8B | 57.8 / 61.6 | 44.3 / 46.2 | 40.3 / 42.1 | 47.5 / 50.0 | - | 46.3 |
| LLaVA-NeXT-QW2 (Liu et al., 2024a) | 7B | 58.0 / - | 47.0 / - | 43.4 / - | 49.5 / - | - | - |
| MiniCPM-V2.6 (Yao et al., 2024b) | 7B | 61.1 / 63.8 | 50.3 / 50.2 | 46.4 / 45.4 | 52.6 / 53.1 | 51.2 | 55.4 |
| LongVU (Shen et al., 2024) | 7B | 64.7 / - | 58.2 / - | 59.5 / - | 60.6 / - | - | 65.4 |
| BOLT (Liu et al., 2025b) | 7B | 66.8 / - | 54.2 / - | 47.3 / - | 56.1 / - | 55.6 | 63.4 |
| Frame-Voyager (Yu et al., 2025) | 8B | 67.3 / - | 56.3 / - | 48.9 / - | 57.5 / - | - | 65.6 |
| LongVILA$^{256frm}$ (Chen et al., 2024b) | 8B | 61.8 / - | 49.7 / - | 39.7 / - | 50.5 / - | - | - |
| Video-XL$^{256frm}$ (Shu et al., 2025) | 7B | 64.0 / 67.4 | 53.2 / 60.7 | 49.2 / 54.9 | 55.5 / 61.0 | 50.7 | 64.9 |
| LLaVA-NeXT-Video (Zhang et al., 2024c) | 34B | 61.7 / 65.1 | 50.1 / 52.2 | 44.3 / 47.2 | 52.0 / 54.9 | 50.5 | 58.8 |
| VILA (Lin et al., 2024b) | 34B | 70.3 / 73.1 | 58.3 / 62.7 | 51.2 / 55.7 | 58.3 / 61.6 | - | 57.8 |
| InternVL2 (Chen et al., 2024c) | 8B | 62.1 / 63.9 | 48.2 / 48.7 | 45.2 / 44.9 | 51.9 / 52.5 | 52.3 | 54.3 |
| **+ KFC** | 8B | 64.5 / 65.4 | 50.0 / 52.3 | 46.5 / 47.3 | 53.5 / 55.0 | 53.3 | 62.2 |
| **+ Nar-KFC** | 8B | 67.2 / 67.7 | 54.7 / 57.9 | 47.1 / 48.9 | **56.3 / 58.1** | **53.9** | **64.4** |
| Qwen2.5-VL (Bai et al., 2025b) | 7B | 65.9 / 66.4 | 54.4 / 54.3 | 45.8 / 46.9 | 55.4 / 55.9 | 52.7 | 55.8 |
| **+ KFC** | 7B | 68.8 / 70.7 | 52.6 / 54.9 | 49.3 / 51.4 | 56.9 / **59.0** | 54.3 | 62.6 |
| **+ Nar-KFC** | 7B | 70.1 / 71.0 | 54.4 / 55.2 | 49.0 / 49.4 | **57.9** / 58.6 | **55.3** | **64.4** |
| LLaVA-OneVision (Li et al., 2024b) | 7B | 65.2 / 67.1 | 51.7 / 54.4 | 45.1 / 46.1 | 53.3 / 55.9 | 54.5 | 58.5 |
| **+ KFC** | 7B | 66.4 / 69.1 | 52.9 / 56.8 | 46.8 / 48.8 | 55.4 / 58.2 | 55.6 | 65.0 |
| **+ Nar-KFC** | 7B | 67.2 / 68.6 | 57.1 / 59.8 | 49.1 / 51.0 | **57.8 / 59.8** | **56.5** | **66.2** |
| LLaVA-Video (Zhang et al., 2024d) | 7B | 67.2 / 69.4 | 53.2 / 53.4 | 47.2 / 47.3 | 55.9 / 56.7 | 54.2 | 60.5 |
| **+ KFC** | 7B | 68.3 / 70.0 | 55.1 / 57.4 | 49.4 / 51.6 | 57.6 / 59.7 | 56.5 | 66.9 |
| **+ Nar-KFC** | 7B | 71.2 / 72.7 | 61.4 / 62.3 | 52.0 / 53.9 | **61.6 / 63.0** | **57.7** | **67.7** |
| InternVL3 (Zhu et al., 2025) | 8B | 68.7 / 70.9 | 58.3 / 58.2 | 50.0 / 50.9 | 59.0 / 60.0 | 53.6 | 60.9 |
| **+ KFC** | 8B | 70.9 / 71.9 | 60.6 / 60.1 | 50.9 / 51.8 | 60.8 / 61.4 | 54.5 | 67.5 |
| **+ Nar-KFC** | 8B | 72.9 / 73.9 | 62.9 / 62.7 | 55.7 / 55.8 | **63.8 / 64.1** | **54.8** | **68.4** |

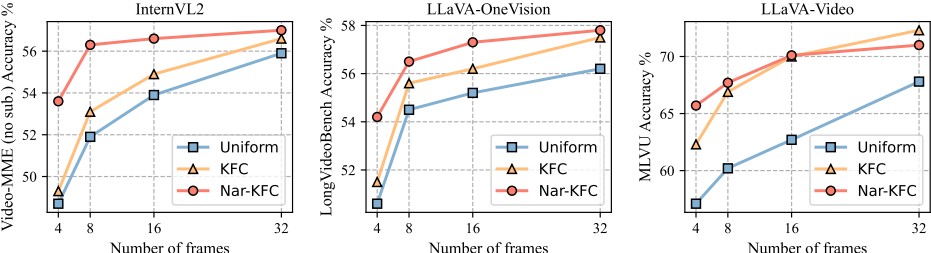

Figure 3: Accuracies (%) of uniform sampling, KFC, and Nar-KFC versus numbers of keyframes.

coverage via interleaved textual narratives. As the number of keyframes increases, the performance gap between uniform sampling and our methods narrows. This can be attributed to: 1) uniform sampling is more likely to capture key moments when more frames are used; and 2) many video QA questions typically only require a few number of frames to accurately answer in current benchmarks. Interestingly, on MLVU, KFC alone outperforms Nar-KFC with 32 keyframes, suggesting that when sufficient keyframes are present, the added benefit of narratives diminishes. These results underscore the strength of KFC in selecting informative keyframes while demonstrating that narratives are particularly valuable when MLLMs have limited context capacity. We further scale Nar-KFC to 72B models and compare them with proprietary models and SOTA VideoLLMs in Appendix §D.2.

**Improvements on open-ended generation tasks.** In Tab. 2), we show Nar-KFC consistently improves performance on open-ended generation tasks that require fine-grained reasoning, demonstrating that even with global-level frame selection and captioning, our methods can still enhance fine-grained tasks. Interestingly, we find that KFC shows decreased performance on the MLVU-OpenEnded summary task, likely because uniformly sampled frames cover the entire video range, whereas KFC-selected frames may be more concentrated. Our Nar-KFC addresses this issue by providing more comprehensive video information.

### 4.3 ABLATION AND ANALYSIS

**KFC and Nar-KFC ablations.** We report the ablation results of KFC and Nar-KFC components on the Video-MME (sub.) and MLVU benchmarks in Tab. 3. Simply inserting narratives between

Table 2: Improvements on open-ended generation tasks. All results are reported based on 8-frame evaluation. For MMBench-Video, `GPT-4-1106` is used as the judge model, while for MLVU-OpenEnded, `GPT-4-0125` serves as the default judge model, following the official implementations.

| Model | MMBench-Video | | | MLVU-OpenEnded | | |
|---|---|---|---|---|---|---|
| | Perception | Reasoning | Overall | Sub_scene | Summary | G-Avg |
| InternVL3-8B (Zhu et al., 2025) | 1.54 | 1.61 | 1.57 | 5.47 | **4.40** | 4.92 |
| **+ KFC** | 1.56 | 1.58 | 1.58 | **5.73** | 4.23 | 4.95 |
| **+ Nar-KFC** | **1.76** | **1.78** | **1.78** | 5.69 | 4.39 | **5.02** |
| Qwen3-VL-8B (Bai et al., 2025a) | 1.62 | 1.65 | 1.64 | 6.17 | **6.01** | 6.09 |
| **+ KFC** | 1.65 | 1.75 | 1.69 | **6.32** | 5.71 | 6.00 |
| **+ Nar-KFC** | **1.75** | **1.76** | **1.76** | 6.28 | 5.97 | **6.12** |

Table 3: Main component ablation results in Nar-KFC. "S, M, L" refer to short, medium, and long video categories in the Video-MME (sub.) benchmark.

| Strategy | Video-MME | | | | MLVU | Time |
|---|---|---|---|---|---|---|
| | S | M | L | Overall | | |
| Uniform | 63.9 | 48.7 | 44.9 | 52.5 | 54.3 | $\mathcal{O}(1)$ |
| + **Nar**ratives | 66.1 | 54.9 | 45.2 | 55.4 | 59.4 | $\mathcal{O}(N)$ |
| **KFC** (IQP) | 65.9 | 52.9 | 46.4 | 55.1 | 62.0 | $\mathcal{O}(2^N)$ |
| **KFC** (GS) | 65.4 | 52.3 | 47.3 | 55.0 | 62.2 | $\mathcal{O}(NK)$ |
| w/o $\mathcal{S}_{\text{QR}}$ | 62.3 | 47.8 | 45.3 | 51.8 | 57.3 | $\mathcal{O}(NK)$ |
| w/o $\mathcal{S}_{\text{FD}}$ | 63.6 | 49.4 | 44.6 | 52.5 | 60.9 | $\mathcal{O}(NK)$ |
| **Nar-KFC** | **67.7** | **57.9** | **48.9** | **58.1** | **64.4** | $\mathcal{O}(NK)$ |

Table 4: Effects of including pre-processing and refinement stages in the KFC Greedy Search (GS) method. V-MME denotes the overall Video-MME (sub.). Line (ii′) indicates Downsampling *without* LowRank. The final KFC (GS) strategy integrates all components from (i) to (iv).

| Ex# | Strategy | V-MME | MLVU |
|---|---|---|---|
| | Vanilla GS | 52.3 | 60.4 |
| (i) | + Initialization | 53.3 | 61.0 |
| (ii) | + LowRank | 53.7 | 61.8 |
| (ii′) | + Downsample | 53.9 | 61.6 |
| (iii) | + LowRank + Downsample | 54.7 | **62.2** |
| (iv) | + Refinement (**KFC**) | **55.0** | 62.2 |

uniformly sampled frames yields improvements of 2.9% on Video-MME and 5.1% on MLVU, indicating that adding narrative context, despite with frames not being query-specific, can effectively boost overall video understanding. To retrieve query-relevant and diverse keyframes, our Greedy Search (GS) strategy achieves results comparable to the optimal Integer Quadratic Programming (IQP) method (55.0% *vs.* 55.1% on Video-MME and 62.2% *vs.* 62.0% on MLVU), while being significantly more efficient with $\mathcal{O}(NK)$ complexity. Details of our IQP implementation and comparisons with GS are provided in Appendix §E.2. Further ablations show that removing the query-relevance score $\mathcal{S}_{\text{QR}}$ leads to a 3.2% drop on Video-MME and 4.9% on MLVU with greedy search. This emphasizes that retrieving query-relevant frames is critical in long videoQA. Meanwhile, incorporating frame diversity $\mathcal{S}_{\text{FD}}$ further stabilizes and enhances performance across benchmarks. When threading all keyframes with interleaved narratives, Nar-KFC achieves the best overall results on all metrics, underscoring its solid effectiveness in representing long video contents.

**Component analysis of greedy search (GS).** Starting from the vanilla GS, which iteratively selects the frame with the highest cumulative score relative to the already selected frames, we progressively incorporate several techniques (Tab. 4) to enhance its effectiveness to a near-optimal solution: (i) initialization with the frame most relevant to the query brings a modest yet consistent gain (from 52.3%→53.3% on Video-MME, and 60.4%→61.0% on MLVU); (ii and iii) applying low-rank denoising and downsampling further improves performance by producing a more compact and less noisy score matrix $\mathbf{S}$; and (iv) adding the final refinement step, KFC (GS) achieves the best results of 55.0% on Video-MME and 62.2% on MLVU. This highlights the cumulative benefit of combining compact frame representations, reduced redundancy, and an iterative selection mechanism.

**Comparisons with other keyframe selection methods.** We compare KFC with several keyframe extraction baselines in Tab. 5, all utilizing the InternVL2 backbone and 8 frames. Details are in Appendix §E.4. Methods that apply top-K frame-query matching using SigLIP (Zhai et al., 2023), or BLIP-2 (Li et al., 2023a) embeddings perform worse than uniform sampling, possibly due to keyframes being concentrated within a narrow temporal window. For those localize-then-answer methods, i.e., TempGQA (Xiao et al., 2024) and SeViLA (Yu et al., 2023), performance heavily depends on the quality of segment localization, which can be unreliable. Recent approaches including DPP (Sun et al., 2025), AKS (Tang et al., 2025), and BOLT (Liu et al., 2025b) generally yield better results by incorporating frame diversity. However, these methods rely on handcrafted and heuristic sampling strategies, lacking

Table 5: Comparisons with different frame selection methods on Video-MME.

| | V-MME (no sub./sub.) |
|---|---|
| InternVL2 | 51.9 / 52.5 |
| + CLIP (top-K) | 47.7 / 50.0 |
| + SigLIP (top-K) | 47.3 / 51.0 |
| + BLIP-2 (top-K) | 47.8 / 50.9 |
| + TempGQA | 50.4 / 51.1 |
| + SeViLA | 52.2 / 53.7 |
| + DPP | 52.2 / 53.5 |
| + AKS | 52.8 / 53.9 |
| + BOLT | 53.3 / - |
| + KFC (Ours) | **53.5 / 55.0** |

Table 6: Analysis of video input components on Video-MME (no sub). Superscript numbers indicate the quantity. Average time and tokens per video are reported.

| Components | V-MME | Latency (s) | TFLOPs ↓ | Token# |
|---|---|---|---|---|
| Narratives[210] | 51.1 | 0.98 | 109.6 | 4,725 |
| Frames[8] (uniform) | 51.9 | 1.03 | 146.3 | 6,280 |
| Frames[8] (KFC) | 53.5 | 1.31 | 146.3 | 6,280 |
| Interleave[8+210] (Nar-KFC) | 56.3 | 2.13 | 202.6 | 11,005 |

Table 7: Temporal structure analysis between narratives and keyframes on Video-MME (no sub) benchmark.

| Temporal Structure | V-MME |
|---|---|
| {Narrative}→{Keyframe}→{Query} | 55.5 |
| {Keyframe}→{Narrative}→{Query} | 55.3 |
| Interleave (Nar-KFC)→{Query} | 56.3 |

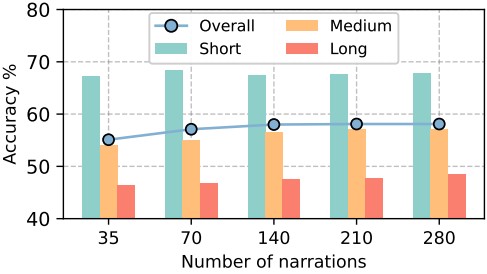

Figure 4: Effect of the total number of inserted narratives, corresponding to the narrative interval △, across videos of different lengths.

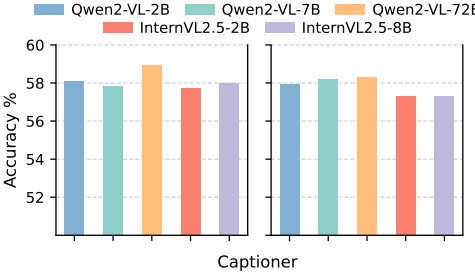

Figure 5: Impact of different captioners for generating narratives. Video-MME (sub.) results are for InternVL2-8B (left) and Qwen2-VL-7B (right).

a principled and generalized frame selection guidance. In comparison, our proposed KFC consistently outperforms all baselines, demonstrating clear superiority in subset frame selection.

**Effect of narrative quantity.** Due to the varying length of videos, we do not directly ablate the effect on a fixed interval value △. Instead, we control the total number of narratives appended, as shown Fig. 4 on Video-MME (sub.). Narratives are incrementally added across 7 intervals between 8 keyframes. The overall accuracy improves steadily from 55.1% to 58.1% as more narratives are available, with more performance gains on medium and long videos. However, since adjacent frames often contain similar visual information, adding more narratives results in diminishing returns due to redundant descriptions. We thus use 210 narratives as the default.

**Effect of narrative quality.** Fig. 5 presents the impact of different captioners on the quality of generated narratives and the resulting performance of Nar-KFC on the Video-MME (sub.). We evaluate five MLLMs of varying sizes and sources as captioners. Narratives extracted from the largest captioner, Qwen2-VL-72B, achieves the best accuracy, i.e., 58.9% on InternVL2-8B and 58.3% on Qwen2-VL-7B, highlighting the benefit of higher-quality narratives. Nevertheless, the overall performance gap across all captioners is small (less than 1%). This suggests that keyframes play a dominant role in long video understanding, while captions serve as auxiliary and supportive context. We thus use the lightweight Qwen2-VL-2B as the default captioner for other benchmarks.

**Efficiency and effectiveness between narratives and keyframes.** We decompose Nar-KFC into standalone narratives and frames in Tab. 6. Although translated from 210 frames, pure narratives perform worse than even 8 uniformly sampled frames (51.1% *vs.* 51.9%), which reflects that substantial information is lost during the frame-to-caption conversion. Nevertheless, narratives exhibit advantages with the shortest latency (0.98s) and the fewest tokens (4,725 per video). Combining narratives with KFC-selected keyframes (Nar-KFC) achieves both the best accuracy and also maintains reasonable efficiency. We discuss detailed computational overhead in Appendix §D.5. In addition, Tab. 7 investigates the temporal structure between narratives and keyframes. Placing all keyframes either before or after the narratives degrades the performance by 0.8% and 1.0%, likely due to disrupted temporal sequences. In contrast, interleaving narratives and frames, as in Nar-KFC, yields superior results. These findings further validate our primary goal: constructing temporally continuous representations for long video understanding.

### 4.4 QUALITATIVE RESULTS

Fig. 6 presents two qualitative examples of our method. In the first example (left), our KFC effectively identifies frames that are both query-relevant and content-diverse, resulting in the correct answer. In the second example (right), we demonstrate that Nar-KFC substantially improves reasoning in a complete relay race scenario by threading temporally interleaved keyframes with coherent narratives.

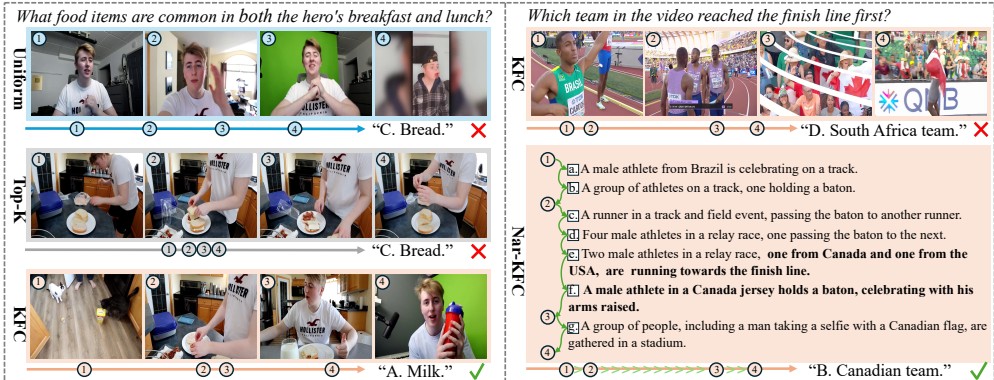

Figure 6: Qualitative results. (left) Comparison of frames selected by uniform sampling, top-K sampling, and our KFC. (right) Key narratives generated by Nar-KFC that lead to the correct answer. Zoom in for details.

This enables accurate inference of the final winner, whereas KFC fails due to limited number of frames. More examples can be found in Appendix §F.

## 5 CONCLUSION

In this paper, we propose a keyframe capturing strategy (KFC) and a narrating keyframe method (Nar-KFC) to boost existing MLLMs for long video understanding, under the constraint of limited context length in language models. Our approach constructs long video representations that are query-relevant, content-diverse, and temporally continuous, all achieved in a training-free manner. This significantly improves the performance of current MLLMs on widely-used long video benchmarks. Our findings strongly validate the potential of MLLMs as effective long video comprehenders.

### ACKNOWLEDGMENTS

This work was supported by Strategic Research Grants from City University of Hong Kong (Project. No. 7005840).

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
