# APPENDIX

## A    LIMITATIONS AND FUTURE WORK

We discuss limitations and possible extensions of Nar-KFC. Despite current MLLMs being able to process our interleaved inputs of keyframes and narratives, thanks to their instruction tuning step, they are not trained with such input formats. This may weaken their ability to fully understand the structure and relationships within our specialized long video representations. A valuable future direction is to incorporate keyframe selection and narrative interleaving into the training of MLLMs, thereby aligning training and testing procedures for improved long video understanding. Furthermore, our method relies mainly on interleaving visual information with narrations and does not incorporate additional modalities such as audio or subtitles. Exploring these modalities in future work may further improve multi-modal long video understanding.

## B    THE USE OF LARGE LANGUAGE MODELS (LLMs)

In this paper, we exclusively utilize advanced LLMs to refine and polish the manuscript. Our prompts to the LLMs include requests such as: "`Please help me polish this academic writing paragraph.  It should be concise, fluent, logical, and in line with academic standards.`" LLMs are not employed for any purposes beyond writing improvement.

## C    BROADER IMPACTS

Effective and efficient long video understanding is a critical task, especially as Internet video streams often last tens of minutes or even hours. We expect that the proposed keyframe selection and narration methods will benefit society by enabling MLLMs to comprehend long videos more accurately and efficiently. However, it is essential to ensure that the narratives generated by specific models remain free from harmful or unrelated content.

## D    MAIN RESULTS SUPPLEMENTARY

We provide supplementary results to the main experiments: Sec. D.1 covers earlier works. Sec. D.2 scales Nar-KFC to 72B models and compares its performance with proprietary models and VideoLLMs capable of reasoning over thousands of frames. Sec. D.3 presents the performance of KFC and Nar-KFC on additional EgoSchema and NExTQA benchmarks, and Sec. D.4 provides a detailed analysis on the MLVU benchmark. Finally, Sec. D.5 discusses the detailed computational overhead introduced by Nar-KFC.

### D.1    COMPREHENSIVE COMPARISONS WITH PREVIOUS METHODS.

VideoLLMs for video understanding have become a popular research area in recent years. However, directly applying previous VideoLLMs to long videos, such as Video-MME, LongVideoBench, and MLVU, often leads to unsatisfactory performance. To provide a more comprehensive comparison, as an extension to the main paper in Tab. 1, we also include the performance of earlier works, such as Video-LLaVA (Lin et al., 2024a), Qwen-VL-Chat (Bai et al., 2023), ST-LLM (Liu et al., 2024d), VideoChat2 (Li et al., 2023b), ShareGPT4Video (Chen et al., 2024a), Chat-UniVi-V1.5 (Jin et al., 2024), and VideoLLaMA2 (Cheng et al., 2024), in Tab. 8.

### D.2    SCALING NAR-KFC TO 72B MODELS

To further evaluate the ability of Nar-KFC to enhance SOTA performance, we scale our Nar-KFC framework to two advanced models: LLaVA-OneVision-72B (32 frames) and LLaVA-Video-72B-Qwen2 (64 frames). We also compare our results with those of SOTA proprietary models and recent works, as shown in Tab. 9 and Tab. 10, with our results highlighted in bold. Extensive experiments demonstrate that the Nar-KFC framework enables 72B models to achieve competitive performance on

Table 8: Comprehensive comparisons with previous VideoLLMs/MLLMs on three common long-video benchmarks: Video-MME, LVB, and MLVU. The reported results are accuracy percentage.

| Model | Size | Video-MME (no sub. / sub.) | | | | LVB $\sim 12m$ | MLVU $\sim 12m$ |
|---|---|---|---|---|---|---|---|
| | | Short | Medium | Long | Overall$_{\sim 17m}$ | | |
| Video-LLaVA (Lin et al., 2024a) | 7B | 45.3 / 46.1 | 38.0 / 40.7 | 36.2 / 38.1 | 39.9 / 41.6 | 39.1 | 47.3 |
| Qwen-VL-Chat (Bai et al., 2023) | 7B | 46.9 / 47.3 | 38.7 / 40.4 | 37.8 / 37.9 | 41.1 / 41.9 | - | - |
| ST-LLM (Liu et al., 2024d) | 7B | 45.7 / 48.4 | 36.8 / 41.4 | 31.3 / 36.9 | 37.9 / 42.3 | - | - |
| VideoChat2 (Li et al., 2023b) | 7B | 48.3 / 52.8 | 37.0 / 39.4 | 33.2 / 39.2 | 39.5 / 43.8 | 39.3 | 44.5 |
| ShareGPT4Video (Chen et al., 2024a) | 8B | 48.3 / - | 36.3 / - | 35.0 / - | 39.9 / - | 41.8 | 46.4 |
| Chat-UniVi-V1.5 (Jin et al., 2024) | 7B | 45.7 / 51.2 | 40.3 / 44.6 | 35.8 / 41.8 | 40.6 / 45.9 | - | - |
| VideoLLaMA2 (Cheng et al., 2024) | 7B | 56.0 / - | 45.4 / - | 42.1 / - | 47.9 / - | - | - |
| VILA (Lin et al., 2024b) | 8B | 57.8 / 61.6 | 44.3 / 46.2 | 40.3 / 42.1 | 47.5 / 50.0 | - | 46.3 |
| LLaVA-NeXT-QW2 (Liu et al., 2024a) | 7B | 58.0 / - | 47.0 / - | 43.4 / - | 49.5 / - | - | - |
| MiniCPM-V-2.6 (Yao et al., 2024b) | 7B | 61.1 / 63.8 | 50.3 / 50.2 | 46.4 / 45.4 | 52.6 / 53.1 | 51.2 | 55.4 |
| LongVU (Shen et al., 2024) | 7B | 64.7 / - | 58.2 / - | 59.5 / - | 60.6 / - | - | 65.4 |
| Frame-Voyager (Yu et al., 2025) | 8B | 67.3 / - | 56.3 / - | 48.9 / - | 57.5 / - | - | 65.6 |
| LongVILA$^{256frm}$ (Chen et al., 2024b) | 8B | 61.8 / - | 49.7 / - | 39.7 / - | 50.5 / - | - | - |
| Video-XL$^{256frm}$ (Shu et al., 2025) | 7B | 64.0 / 67.4 | 53.2 / 60.7 | 49.2 / 54.9 | 55.5 / 61.0 | 50.7 | 64.9 |
| VILA (Lin et al., 2024b) | 34B | 70.3 / 73.1 | 58.3 / 62.7 | 51.2 / 55.7 | 58.3 / 61.6 | - | 57.8 |
| InternVL2 (Chen et al., 2024c) | 8B | 62.1 / 63.9 | 48.2 / 48.7 | 45.2 / 44.9 | 51.9 / 52.5 | 52.3 | 54.3 |
| **+ KFC** | 8B | 64.3 / 65.4 | 49.6 / 52.3 | 46.1 / 47.3 | 53.1 / 55.0 | **53.3** | 62.2 |
| **+ Nar-KFC** | 8B | 67.2 / 67.7 | 54.7 / 57.9 | 47.1 / 48.9 | **56.3 / 58.1** | **53.9** | **64.4** |
| Qwen2-VL (Wang et al., 2024b) | 7B | 65.7 / 66.9 | 52.8 / 53.0 | 46.7 / 48.6 | 55.0 / 56.1 | 53.4 | 59.6 |
| **+ KFC** | 7B | 68.2 / 69.7 | 53.3 / 54.9 | 48.4 / 50.2 | **56.7 / 58.3** | **54.6** | 65.9 |
| **+ Nar-KFC** | 7B | 68.8 / 69.3 | 53.4 / 55.3 | 48.0 / 49.0 | **56.7 / 57.9** | 53.6 | **68.5** |
| Qwen2.5-VL (Bai et al., 2025b) | 7B | 65.9 / 66.4 | 54.4 / 54.3 | 45.8 / 46.9 | 55.4 / 55.9 | 52.7 | 55.8 |
| **+ KFC** | 7B | 68.8 / 70.7 | 52.6 / 54.9 | 49.3 / 51.4 | 56.9 / **59.0** | 54.3 | 62.6 |
| **+ Nar-KFC** | 7B | 70.1 / 71.0 | 54.4 / 55.2 | 49.0 / 49.4 | **57.9 / 58.6** | **55.3** | **64.4** |
| LLaVA-OneVision (Li et al., 2024b) | 7B | 65.2 / 67.1 | 51.7 / 54.4 | 45.1 / 46.1 | 53.3 / 55.9 | 54.5 | 58.5 |
| **+ KFC** | 7B | 66.4 / 69.1 | 52.9 / 56.8 | 46.8 / 48.8 | 55.4 / 58.2 | 55.6 | 65.0 |
| **+ Nar-KFC** | 7B | 67.2 / 68.6 | 57.1 / 59.8 | 49.1 / 51.0 | **57.8 / 59.8** | **56.5** | **66.2** |
| LLaVA-Video (Zhang et al., 2024d) | 7B | 67.2 / 69.4 | 53.2 / 53.4 | 47.2 / 47.3 | 55.9 / 56.7 | 54.2 | 60.5 |
| **+ KFC** | 7B | 68.3 / 70.0 | 55.1 / 57.4 | 49.4 / 51.6 | 57.6 / 59.7 | 56.5 | 66.9 |
| **+ Nar-KFC** | 7B | 71.2 / 72.7 | 61.4 / 62.3 | 52.0 / 53.9 | **61.6 / 63.0** | **57.7** | **67.7** |
| InternVL3 (Zhu et al., 2025) | 8B | 68.7 / 70.9 | 58.3 / 58.2 | 50.0 / 50.9 | 59.0 / 60.0 | 53.6 | 60.9 |
| **+ KFC** | 8B | 70.9 / 71.9 | 60.6 / 60.1 | 50.9 / 51.8 | 60.8 / 61.4 | 54.5 | 67.5 |
| **+ Nar-KFC** | 8B | 72.9 / 73.9 | 62.9 / 62.7 | 55.7 / 55.8 | **63.8 / 64.1** | **54.8** | **68.4** |
| Qwen3-VL (Team, 2025) | 8B | 68.4 / 70.7 | 55.4 / 55.3 | 50.1 / 52.0 | 58.0 / 59.1 | 54.7 | 49.5 |
| **+ KFC** | 8B | 68.4 / 71.9 | 57.3 / 57.0 | 50.8 / 50.7 | 58.9 / 59.9 | 55.8 | 63.0 |
| **+ Nar-KFC** | 8B | 70.4 / 72.9 | 60.1 / 59.7 | 52.7 / 52.4 | **61.1 / 61.7** | **56.2** | **65.8** |

Table 9: Scaling to 72B models on the Video-MME benchmark. Results from our Nar-KFC method are in bold.

| Model | Frames | Video-MME (no sub.) | | | |
|---|---|---|---|---|---|
| | | Short | Medium | Long | Overall |
| LLaVA-OneVision-72B | 32 | 76.7 | 62.2 | 60.0 | 66.3 |
| + **Nar-KFC** | 32 | **77.5** | **68.6** | **61.9** | **69.6** |
| LLaVA-Video-72B | 64 | 81.7 | 67.9 | 61.8 | 70.4 |
| + **Nar-KFC** | 64 | **82.0** | **68.9** | **63.6** | **71.5** |
| VideoChat-Flash@448-7B (Li et al., 2024c) | N/A | - | - | - | 65.3 |
| LLaVA-OneVision-72B + T* (Ye et al., 2025) | 32 | 77.5 | 66.6 | 61.0 | 68.3 |
| VILAMP-7B (Cheng et al., 2025a) | 1 fps | - | - | - | 67.5 |
| Aria-8x3.5B | 256 | 76.9 | 67.0 | 58.8 | 67.6 |
| GPT-4o (0615) | 384 | 80.0 | 70.3 | 65.3 | 71.9 |
| Qwen2-VL-72B (Wang et al., 2024b) | 768 | 80.1 | 71.3 | 62.2 | 71.2 |
| AdaReTake-72B (Wang et al., 2025a) | 2 fps | - | - | - | 73.5 |
| Gemini-1.5-Pro (0615) | 1/0.5 fps | 81.7 | 74.3 | 67.4 | 75.0 |

Table 10: Scaling to 72B models on the MLVU benchmark. Results from our Nar-KFC method are shown in bold. * indicates results obtained from our own implementation.

| Model | Frames | MLVU |
|---|---|---|
| LLaVA-OneVision-72B | 32 | 66.4 |
| + **Nar-KFC** | 32 | **74.4** |
| LLaVA-Video-72B | 64 | 74.4 (73.6*) |
| + **Nar-KFC** | 64 | **75.0** |
| GPT-4o (0615) | 0.5 fps | 64.6 |
| VideoLLaMA3-7B (Zhang et al., 2025) | ≤180 | 73.0 |
| VILAMP-7B (Cheng et al., 2025a) | 1 fps | 72.6 |
| VideoChat-Flash@448-7B (Li et al., 2024c) | 1 fps | 74.7 |
| AdaReTake-72B (Wang et al., 2025a) | 2 fps | 78.1 |

the Video-MME benchmark (71.5%) and leading results on MLVU (75.0%). Notably, our approach uses significantly fewer frames (32 or 64) compared to proprietary models such as Gemini-1.5-Pro and VideoLLMs that reason over thousands of frames, including VILAMP (Cheng et al., 2025a) and AdaReTake (Wang et al., 2025a). These findings underscore the potential significance of our framework, particularly under the limited context length constraints of MLLMs.

## D.3  RESULTS ON MORE BENCHMARKS

Table 11: Results on EgoSchema and NExTQA benchmarks. Accuracy sign % is omitted for clarity.

| Model | Frames | EgoSchema 3min | NExT-QA 0.7min |
|---|---|---|---|
| InternVideo (Wang et al., 2022) | 90 | 32.1 | 49.1 |
| LLoVi (Zhang et al., 2024a) | 90 | 57.6 | 67.7 |
| LangRepo (Kahatapitiya et al., 2024) | 180 | 66.2 | 60.9 |
| VideoAgent (Wang et al., 2024d) | 8.4 | 60.2 | 71.3 |
| LVNet (Park et al., 2024) | 12 | 66.0 | 72.9 |
| VidF4 (Liang et al., 2024) | 8 | - | 74.1 |
| VideoTree (Wang et al., 2025b) | 63.2 | 66.2 | 73.5 |
| InternVL2-8B (Chen et al., 2024c) | | 59.8 | 76.5 |
| + **KFC** | 8 | 58.6 | 77.8 |
| + **Nar-KFC** | | **64.0** | **78.1** |
| Qwen2-VL-7B (Wang et al., 2024b) | | 60.8 | 76.3 |
| + **KFC** | 8 | 63.2 | 76.6 |
| + **Nar-KFC** | | **65.8** | **77.6** |

We further report performance of our KFC and Nar-KFC on two relatively shorter video benchmarks, i.e., EgoSchema (Subset) (Mangalam et al., 2023) and NExTQA (Xiao et al., 2021), in Tab. 11.

Unlike the long video datasets discussed in the main paper, our keyframe selection strategy (i.e., KFC) may underperfom compared to uniform sampling when applied to shorter videos. For example,

InternVL2-8B yields 58.6% accuracy on EgoSchema when using KFC. This performance drop is primarily due to KFC disrupting the temporal consistency of frame sequences, which is particularly important for short video understanding. Nevertheless, supplementing with non-keyframe narratives (Nar-KFC) leads to consistent performance improvements even on these shorter benchmarks. The gains are especially evident on EgoSchema, while the improvement on NExTQA is more limited, likely due to its relatively short average video length of approximately 44 sec.

Table 12: Results on TempCompasss and Video-Holmes benchmarks. Accuracy sign % is omitted for clarity.

| Model | TempCompass | | Video-Holmes |
| | Caption Matching | Overall | |
| --- | --- | --- | --- |
| Qwen2.5-VL-7B (Bai et al., 2025b) | 74.0 | 72.2 | 20.4 |
| + KFC | **74.1** | **72.2** | 20.7 |
| + Nar-KFC | - | - | **22.9** |
| InternVL3-8B (Zhu et al., 2025) | 80.1 | 74.8 | 33.5 |
| + KFC | **80.2** | **74.9** | **34.5** |
| + Nar-KFC | - | - | **34.5** |
| Qwen3-VL-8B (Team, 2025) | 79.1 | **74.4** | 30.9 |
| + KFC | **80.0** | 74.3 | 33.7 |
| + Nar-KFC | - | - | **33.8** |

We also evaluate our methods on extremely short video understanding benchmarks (TempCompass (Liu et al., 2024e), 10s), where narratives are not required, as well as on the more complex video reasoning benchmark, Video-Holmes (Cheng et al., 2025b), as illustrated in Tab. 12. For the relatively short TempCompass benchmark, selecting query-relevant and diversified keyframes results in considerable overlap with uniform sampling, leading to limited performance improvement. In contrast, on the more challenging Video-Holmes benchmark, our approach of carefully selecting keyframes and incorporating threaded narratives significantly enhances the MLLM's video reasoning capabilities.

## D.4 DETAILED ANALYSIS ON MLVU CATEGORIES

In Fig.7, we provide a detailed comparison of performance across specific categories in the MLVU benchmark as a supplement to the main paper Tab. 1. Compared to uniform sampling, the overall performance improvement introduced by KFC across all four models is primarily attributed to its superior accuracy in the **needle** and **count** categories. The *needle* task involves questions based on rare or unusual frames sourced from external videos, which are more likely to be captured by our query-relevance-based sampling strategy. In contrast, such frames are often missed by uniform sampling. A similar challenge arises in the *count* task, where correct answers rely on retrieving specific frames first in order to support accurate object/crowd/event counting.

On the other hand, our Nar-KFC approach generally achieves the best performance on **plotQA** and **topic** tasks. This advantage stems from its ability to preserve temporal continuity, which is often lacking in KFC-optimized keyframes that are temporally sparse and discontinuous. Such discontinuity hinders the model's ability to comprehend holistic video contents. For instance, KFC performs the worst on the *topic* task when inferenced with LLaVA-OneVision (c) and LLaVA-Video (d), even underperforming the uniform sampling baseline. In contrast, Nar-KFC addresses this issue through a narrative threading strategy, which maintains continuity by supplementing keyframes with coherent non-keyframe descriptions. This strategy significantly enhances the model's understanding of overall video plots and topics.

## D.5 COMPUTATIONAL OVERHEAD

We analyze and present the detailed computational complexity (efficiency), including TFLOPs, latency, and memory usage, in Tab. 13. Note that searching the entire space of IQP would require approximately $10^{13}$ TFLOPs, making it impractical in real-world scenarios. Therefore, we report the computational complexity based on using 30k nodes in the IQP algorithm. Here, "search efficiency" refers to the keyframe search stage, while "overall efficiency" primarily pertains to the MLLM reasoning stage.

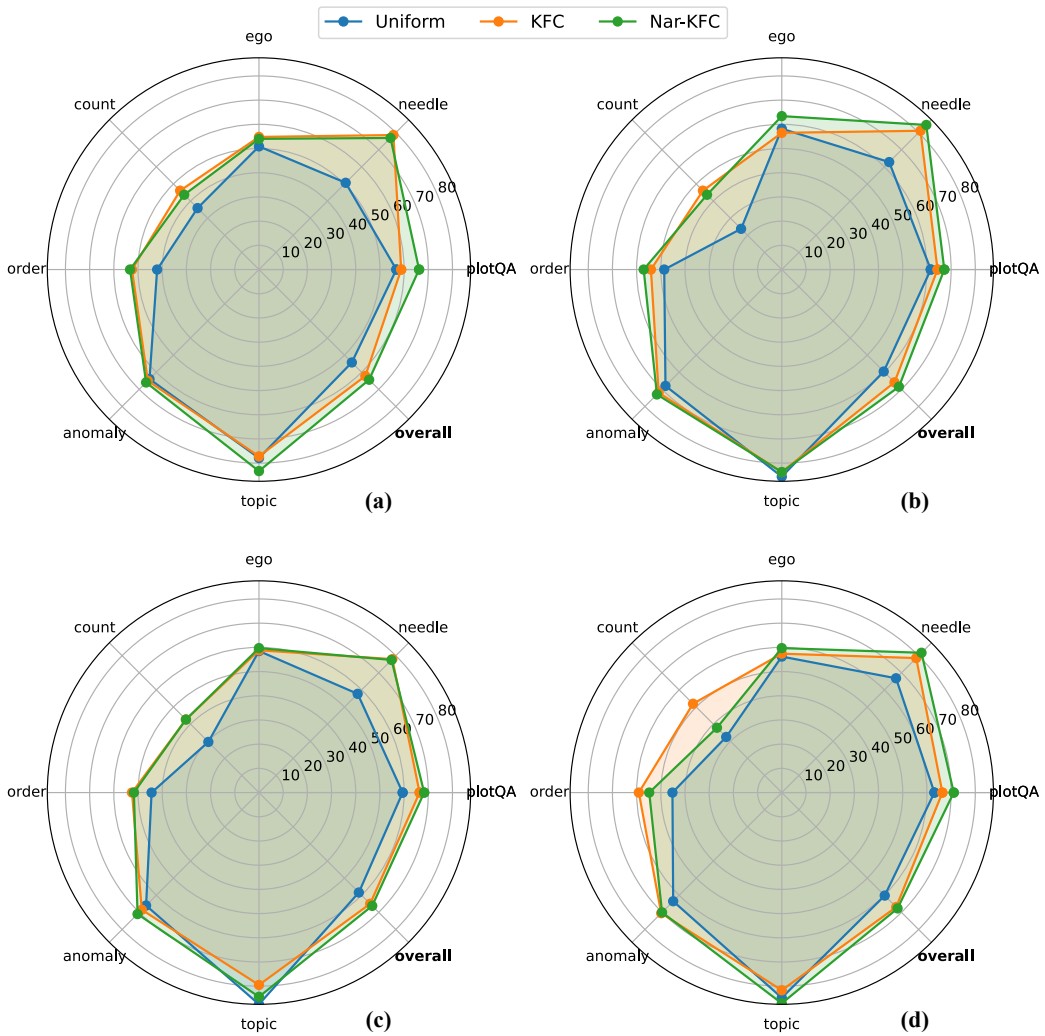

Figure 7: Performance comparison across specific categories of the MLVU benchmark. Results are shown for (a) InternVL2-8B, (b) Qwen2-VL-7B, (c) LLaVA-OneVision-7B, and (d) LLaVA-Video-7B, evaluated using three keyframe selection strategies: Uniform, KFC, and Nar-KFC.

Table 13: Computational efficiency comparison, including TFLOPs, latency, and memory usage for both the searching and overall inference stages. Results are reported using 8 frames and 210 narratives with the InternVL2-8B model.

| Method | Search Efficiency | | | Overall Efficiency | | |
|---|---|---|---|---|---|---|
| | TFLOPs↓ | Latency (s) ↓ | Memory (GB) | TFLOPs↓ | Latency (s) ↓ | Memory (GB) |
| Uniform-8 | N/A | 0.20 | N/A | 146.3 | 1.03 | 21.8 |
| Top-k | N/A | 0.24 | N/A | 146.3 | 1.07 | 21.8 |
| KFC (IQP-30k) | 6.9 | 7.18 | N/A | 153.2 | 8.01 | 21.8 |
| KFC (GS) | ∼0 | 0.48 | N/A | 146.3 | 1.31 | 21.8 |
| Nar-KFC | ∼0 | 0.60 | N/A | 202.6 | 2.13 | 32.2 |

Since we use an offline CLIP model to extract video embeddings (including query embeddings) and a Qwen2-VL-2B captioning model to generate video narratives, we also report their computational complexity in Tab. 14. The results are evaluated on an average 17-minute video (1,020 frames at 1 fps). It is important to note that these extraction processes are performed offline prior to online reasoning, which is the same as all previous keyframe selection strategies. Therefore, although the

preprocessing step is time-consuming, it impacts all keyframe selection methods equally, but does not impact the final inference complexity.

For an on-demand (long) video understanding system and suppose we are given an on-demand video, our lightweight captioner only needs to extract less than 210 narratives no matter how long the video is (since we have proved in our paper that more narrations won't bring further improvements and may exceed the context length of MLLMs). The caption extraction process requires less than 74.2 sec of latency. In practice, there are often no more than 210 frames between the first and last sampled keyframes, which can further reduce preprocessing time. The low computational cost of captioning is primarily due to our lightweight captioner, as we demonstrate that Nar-KFC's performance is not sensitive to captioner size and only a small number of frames are processed. If a latency of 74.2 sec (or less) remains a concern for on-demand video systems, our keyframe selection method, KFC-GS, can be used without the captioning stage for faster inference compared with prior frame selection methods. Overall, our approach achieves a favorable balance between accuracy and efficiency.

Table 14: Computational overhead for CLIP embedding extraction and frame captioning. Results are reported on an average 17 min video at 1 fps (1020 frames) frame sampling.

| Model | Frames | TFLOPs↓ | Latency (s)↓ | Memory (GB) |
|---|---|---|---|---|
| *Offline Frame Embedding & Caption Extraction* | | | | |
| CLIP-ViT-L-336px | 1020 | 420.8 | 25.8 | 1.6 |
| Qwen2-VL-2B | 1020 | 4462.5 | 360.5 | 7.2 |
| *On-demand Video System Processing* | | | | |
| Qwen2-VL-2B | ≤210 | ≤ 918.8 | ≤74.2 | 7.2 |

# E  ADDITIONAL ABLATION RESULTS

## E.1  A SYMMETRICAL FORMULATION OF ORIGINAL OBJECTIVE AND ANALYSIS.

**Objective Revisiting.** In the main paper Sec. 3, we formulate the keyframe selection task as a *graph* problem and model it using integer quadratic programming (IQP) (3). However, the constructed score matrix (1) is asymmetric, as it only accounts for the query relevance of the $i$-th frame and the diversity between the $i$-th and $j$-th frames, while neglecting the query relevance of the $j$-th frame. This asymmetry introduces a minor discrepancy compared to the standard subgraph selection procedure. We illustrate this discrepancy with an example.

**Example.** Suppose we aim to retrieve 3 keyframes from 5 frames, and the optimal selection is given by $\mathbf{x} = [1, 1, 1, 0, 0]^T$, indicating that first three frames are selected. The score matrix $\mathbf{S}$ is defined as:

$$\mathbf{S}_{i,j} = \mathcal{S}(i, j) = S_{\mathrm{QR}}(i) + S_{\mathrm{FD}}(i, j) = \begin{bmatrix} 0 & a_{12} & a_{13} & a_{14} & a_{15} \\ 0 & 0 & a_{23} & a_{24} & a_{25} \\ 0 & 0 & 0 & a_{34} & a_{35} \\ 0 & 0 & 0 & 0 & a_{45} \\ 0 & 0 & 0 & 0 & 0 \end{bmatrix}. \tag{5}$$

where $a_{i,j}$ denotes the score term for $i < j$ (i.e., only the upper trangular part of $\mathbf{S}$ is considered). According to (3), the maximum sum score (the total edge weight of the subgraph) should be:

$$\begin{aligned} \mathbf{x}^T \mathbf{S} \mathbf{x} &= [1, 1, 1, 0, 0] \begin{bmatrix} 0 & a_{12} & a_{13} & a_{14} & a_{15} \\ 0 & 0 & a_{23} & a_{24} & a_{25} \\ 0 & 0 & 0 & a_{34} & a_{35} \\ 0 & 0 & 0 & 0 & a_{45} \\ 0 & 0 & 0 & 0 & 0 \end{bmatrix} [1, 1, 1, 0, 0]^T \\ &= [1, 1, 1, 0, 0][a_{12} + a_{13}, a_{23}, 0, 0, 0]^T \\ &= a_{12} + a_{13} + a_{23} \\ &= S_{\mathrm{QR}}(1) + S_{\mathrm{FD}}(1, 2) + S_{\mathrm{QR}}(1) + S_{\mathrm{FD}}(1, 3) + S_{\mathrm{QR}}(2) + S_{\mathrm{FD}}(2, 3). \end{aligned} \tag{6}$$

From this computation, we know that the query relevance of the first frame is counted twice, while that of the last selected frame ($3^{rd}$) is not counted at all, as there are no subsequent frames after it. This *discrepancy* shows the deviation from the standard graph-based subgraph selection formulation.

**Symmetric Score Matrix.** To mitigate this discrepancy and align the keyframe selection process with a standard graph problem, we reconstruct the original score matrix $\mathbf{S}$ to be symmetric by incorporating the query relevance of the $j$-th frame, defined as:

$$\mathbf{S}_{i,j} = S(i,j) = S_{\mathrm{QR}}(i) + 2S_{\mathrm{FD}}(i,j) + S_{\mathrm{QR}}(j). \tag{7}$$

**Experimental Results and Analysis.** Compared with the symmetric $\mathbf{S}$ in (7), our original asymmetric matrix involves fewer terms with reducing size (only the upper triangular part is calculated), which leads to faster inference. Tab. 15 presents additional experimental results for replacing the original score matrix $\mathbf{S}$ with its symmetric counterpart. Modifying $\mathbf{S}$ to be symmetric – thus aligning the formulation with a standard graph problem – results in a 1% performance drop when using the IQP solver. This result supports the benefit of assigning higher weights to the initially selected frame at the beginning. Since the first keyframe is heuristically selected based on query relevance, this modification has negligible impact when using the GS strategy. We thus adopt the asymmetric score matrix defined in (1) for the remainder of our process.

Table 15: Impact of whether replacing score matrix to its symmetric counterpart. Results are reported on the Video-MME (sub.) benchmark using InternVL2-8B model. The search node number is 40k for solving IQP.

| Setting | Strategy | Video-MME (sub.) | | | |
| --- | --- | --- | --- | --- | --- |
| | | Short | Medium | Long | Overall |
| asymmetric $\mathbf{S}$ (1) | IQP | 65.9 | 52.9 | 46.4 | **55.1** |
| symmetric $\mathbf{S}$ (7) | | 66.1 | 50.1 | 46.1 | 54.1 |
| asymmetric $\mathbf{S}$ (1) | GS | 65.4 | 52.3 | 47.3 | 55.0 |
| symmetric $\mathbf{S}$ (7) | | 65.7 | 52.6 | 47.2 | **55.1** |

### E.2 INTEGER QUADRATIC PROGRAMMING (IQP) *vs.* GREEDY SEARCH (GS)

Table 16: Impact of expanding the IQP search space on performance and efficiency. Results are reported on the Video-MME (sub.) benchmark using InternVL2-8B model, with average computational time per video (in seconds) evaluated on a single NVIDIA A100 GPU.

| Setting | Nodes# | Video-MME (sub.) | | | | Time (s) |
| --- | --- | --- | --- | --- | --- | --- |
| | | Short | Medium | Long | Overall | |
| Uniform | - | 63.9 | 48.7 | 44.9 | 52.5 | 1.03 |
| GS | - | 65.4 | 52.3 | 47.3 | 55.0 | 1.31 |
| IQP | 5k | 64.2 | 52.6 | 46.7 | 54.5 | 3.91 |
| | 10k | 64.4 | 52.6 | 45.8 | 55.0 | 4.81 |
| | 20k | 64.3 | 52.3 | **47.9** | 54.9 | 6.23 |
| | 30k | 65.6 | 52.6 | 46.2 | 54.8 | 8.01 |
| | 40k | **65.9** | **52.9** | 46.4 | **55.1** | 9.26 |
| IQP (GS init) | 5k | 64.3 | 52.0 | 48.0 | 54.7 | 5.22 |
| | 10k | 65.1 | 51.9 | 47.3 | 54.5 | 6.12 |
| | 20k | 65.1 | 52.3 | 47.5 | 54.9 | 7.54 |
| | 30k | 65.3 | 52.3 | 45.7 | 54.4 | 9.32 |
| | 40k | 65.8 | 51.4 | 46.0 | 54.4 | 10.57 |

We implement the Integer Quadratic Programming (IQP) algorithm using CPLEX and set a maximum number of search nodes to obtain the optimal set of keyframe indices within a limited time. The corresponding IQP results are reported in Tab. 16. As the search space increases from 5k to 40k nodes, performance on short videos gradually improves from 64.2% to 65.9%, which validates the effectiveness of modeling keyframe selection as an IQP problem. However, this improvement does not hold for long videos, where performance becomes unstable as the search space expands. We speculate that this is because even 40k nodes are still insufficient to cover the full solution space for long videos. For instance, in a 15-minute video (900 frames at 1 fps), selecting 8 keyframes results

in approximately $C(900, 8) \simeq 2.5 \times 10^{18}$, i.e., roughly 2.5 quintillion possible combinations. This vast search space far exceeds what can be practically explored with a node limit of 40k, let alone for videos that span several hours.

We also attempt to initialize the IQP search with greedy searched results, which are highlighted in gray in Tab. 16, in hopes of better guiding the IQP solving process. Experimental results indicate that this initialization strategy does not lead to further improvements in IQP performance, likely due to the search space remaining too large to be effectively navigated. Therefore, we adopt a customized greedy search (GS) strategy as a practical and robust alternative to the IQP algorithm.

### E.3 ABLATIONS ON HYPERPARAMETERS IN KFC (GS)

Table 17: Impact of low-rank truncation $r$ in our Greedy Seach (GS) algorithm.

| LowRank truncation $r$ | Video-MME (sub.) | | | |
| --- | --- | --- | --- | --- |
| | Short | Meidum | Long | Overall |
| $N/16$ | 64.8 | 51.3 | 46.3 | 54.2 |
| $N/8$ | 65.3 | 51.7 | 46.0 | 54.3 |
| $N/4$ | **65.4** | **52.3** | 47.3 | **55.0** |
| $N/2$ | 65.2 | 51.7 | **47.6** | 54.8 |
| $N$ (w/o SVD) | 64.1 | 51.8 | 45.7 | 53.9 |

The low-rank truncation parameter $r$ in SVD (Sec. 3.1.2) serves to compress and denoise neighboring frames in the score matrix $\mathcal{S}$. Setting $r$ equal to the number of video frames $N$ is equivalent to not applying the SVD technique. Our experiments in Tab. 17 demonstrate that incorporating this decomposition step facilitates frame selection and reduces the problem size. Setting $r = \frac{N}{4}$ yields the best performance, where $N$ refers to the total number of frames in a video. Choosing a smaller value, such as $\frac{N}{16}$ or $\frac{N}{8}$, leads to excessive information loss and consequently degrades the performance.

Table 18: Impact of downsample resolution in our Greedy Seach (GS) algorithm.

| Downsample Resolution | Video-MME (sub.) | | | |
| --- | --- | --- | --- | --- |
| | Short | Meidum | Long | Overall |
| 64 | 63.8 | 52.3 | 44.0 | 53.4 |
| 128 | **65.4** | **52.3** | 47.3 | **55.0** |
| 256 | 64.8 | 50.2 | 47.6 | 54.2 |
| 512 | 63.0 | 51.4 | **47.7** | 53.8 |

Following previous works such as Frame-Voyager (Yu et al., 2025) and MDP3 (Sun et al., 2025), we default to downsampling the frame sequence to 128 frames. Our experiments, as shown in Tab. 18, also indicate that this downsampling resolution generally yields the best performance. Similar to SVD, the downsampling operation is designed to balance the trade-off between denoising the score matrix and minimizing the information loss.

Table 19: Impact of refinement window size $k$ in our Greedy Seach (GS) algorithm.

| Window Size $k$ | Video-MME (sub.) | | | |
| --- | --- | --- | --- | --- |
| | Short | Meidum | Long | Overall |
| 0 (w/o refine) | 65.0 | 51.2 | **47.8** | 54.7 |
| 1 | 65.1 | 51.4 | 47.4 | 54.7 |
| 2 | **65.4** | 52.3 | 47.3 | **55.0** |
| 4 | 64.6 | **53.1** | 45.7 | 54.4 |
| 8 | 64.2 | 50.3 | 43.8 | 52.8 |

We analyze the impact of the neighbor window size $k$ in the final Greedy Search (GS) refinement step. As shown in Tab. 19, setting $k = 0$ corresponds to using the GS strategy without any refinement.

Table 20: Impact of incorporating full video-level narratives. These narratives include segments that appear before the first keyframe and after the last keyframe. $^*$ indicates that only narratives *between keyframes* are utilized in Nar-KFC.

| Setting | Video-MME (no sub. / sub.) | | | |
|---|---|---|---|---|
| | Short | Meidum | Long | Overall |
| Full-Narrative | 66.3 / 66.9 | 56.3 / 58.0 | 46.7 / 47.3 | **56.4** / 57.4 |
| Nar-KFC$^*$ | 67.2 / 67.7 | 54.7 / 57.9 | 47.1 / 48.9 | 56.3 / **58.1** |

When $k = 2$, which means examining a total of four frames, two before and two after the selected keyframe, the model achieves the best overall performance. This highlights the effectiveness of the refinement step as a robust strategy to complement prior SVD and downsampling operations. However, increasing the window size further (e.g., $k = 4$ or $k = 8$) results in performance degradation. This is likely due to the disruption of holistic keyframe combinations constructed by the greedy search, as excessive frame examination may introduce noise or redundancy.

**Conclusion.** The core of KFC-GS is a greedy algorithm, which iteratively selects the next frame with the highest cumulative score. Although we incorporate some pre-processing (SVD, downsampling) and post-processing steps (refinement) to further enhance performance, the vanilla greedy selection (GS) is already highly effective with initialization, eg., achieving 53.3 on Video-MME and 61.0 on MLVU. These results demonstrate that KFC-GS is generally robust and capable of generalizing well across different benchmarks. In fact, we do not manually tune the hyperparameters $(r, d)$ involved in the pre-processing techniques, as they can be empirically set within an appropriate range. Comprehensive ablations on these hyperparameters (Tab. 17, Tab. 18, Tab. 19)further demonstrate that our results are not particularly sensitive to these parameters. Re-tuning is generally unnecessary, as our approach consistently achieves improvements over multiple benchmarks with 4 different MLLMs.

## E.4 Implementation Details of Frame Extraction Baselines in Tab. 5

For CLIP[1] (Radford et al., 2021), SigLIP[2] (Zhai et al., 2023), and BLIP-2[3] (Li et al., 2023a), we directly rank and select the top-K candidate keyframes based on their frame-query cosine similarity logits. For TempGQA (Xiao et al., 2024), we follow the official code[4] to first select a segment based on the question, and then uniformly sample frames from the selected segment to generate the answer. For SeViLA (Yu et al., 2023), we use its trained localizer[5] to select the $K$ keyframes as input, while maintaining the original hyperparameter settings. As for DPP (Determinantal Point Process) selection (Sun et al., 2025), since the official code is unavailable, we reimplement the DPP algorithm by defining its kernel matrix as $\mathcal{S}(i, q)\mathcal{S}(j, q)[1 - \mathcal{S}(i, j)]$, where the first two terms represent the similarity of frames $i, j$ to the query $q$, and the last term encourages frame diversity between frame $i$ and frame $j$. $\mathcal{S}$ denotes the cosine similarity operation. For AKS (Tang et al., 2025), we select keyframes based on the frame scores provided in the official repository[6].

## E.5 Incorporating Full Video-level Narratives.

Our default Nar-KFC configuration (see main paper Sec. 3.2) only uses narratives that appear between the first and the last keyframe, discarding those that occur at the beginning or end of the video. Here, we analyze the effect of incorporating full video-level narratives, as shown in Tab. 20, while keeping the total number of inserted narratives fixed at 210. The results suggest that including these additional narratives has minimal impact on overall video understanding. This finding further supports our primary conclusion: keyframes play a dominant role in long-form VideoQA, while narratives mainly serve as auxiliary context.

---

[1]https://huggingface.co/openai/clip-vit-large-patch14-336
[2]https://huggingface.co/google/siglip-so400m-patch14-384
[3]https://huggingface.co/Salesforce/blip2-opt-2.7b
[4]https://github.com/doc-doc/NExT-GQA/tree/main/code/TempGQA
[5]https://github.com/Yui010206/SeViLA?tab=readme-ov-file
[6]https://github.com/ncTimTang/AKS

## F    ADDITIONAL QUALITATIVE EXAMPLES

We present additional qualitative examples of our keyframe selection method (KFC) in Fig. 8, and of the narrating keyframe method (Nar-KFC) in Fig. 9. Note that the frames leading to incorrect predictions in Fig. 9 can be regarded as failure cases of KFC.

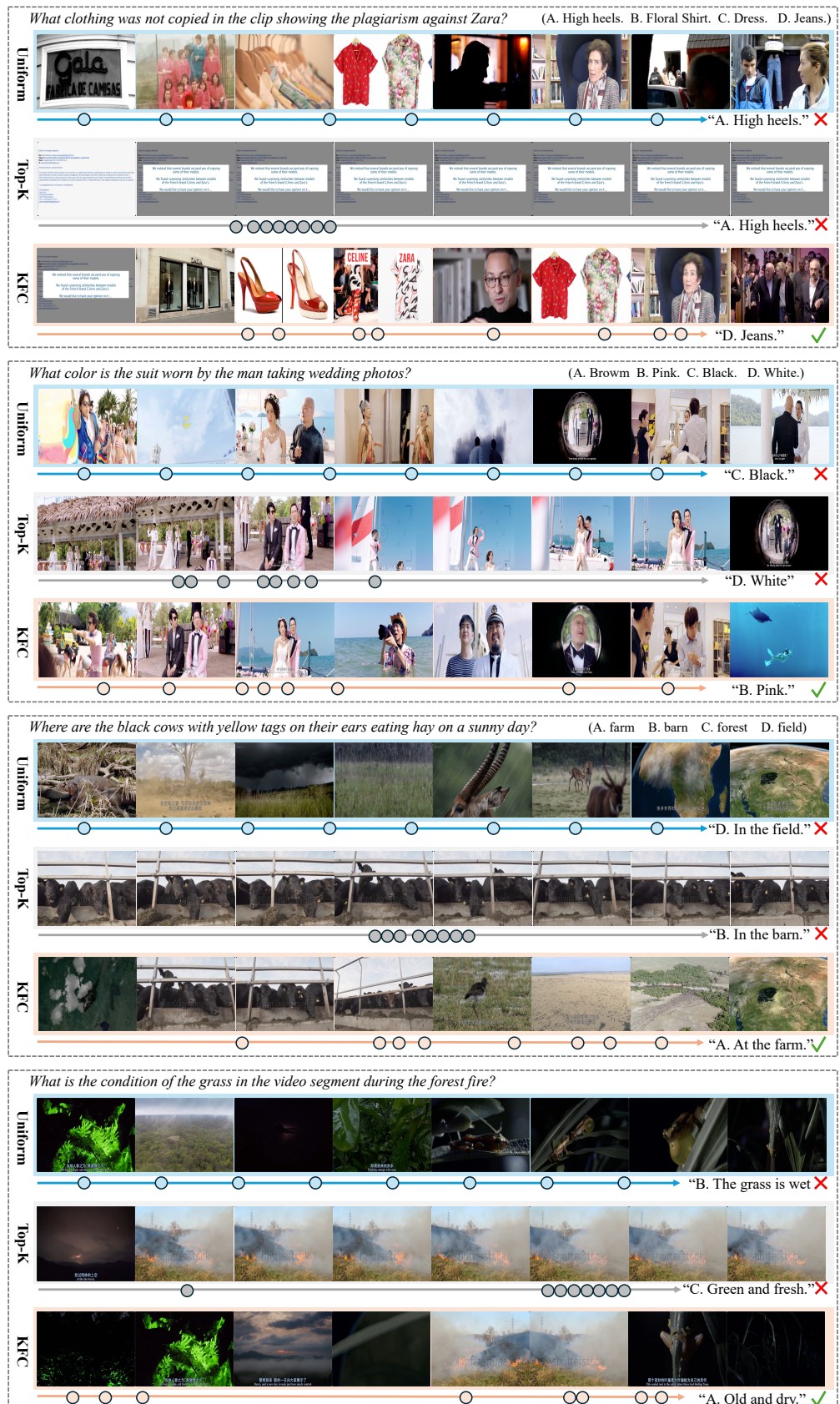

Figure 8: More qualitative examples of keyframe selection using our KFC method, compared with uniform sampling and topK sampling baselines. Zoom in for better visual details.

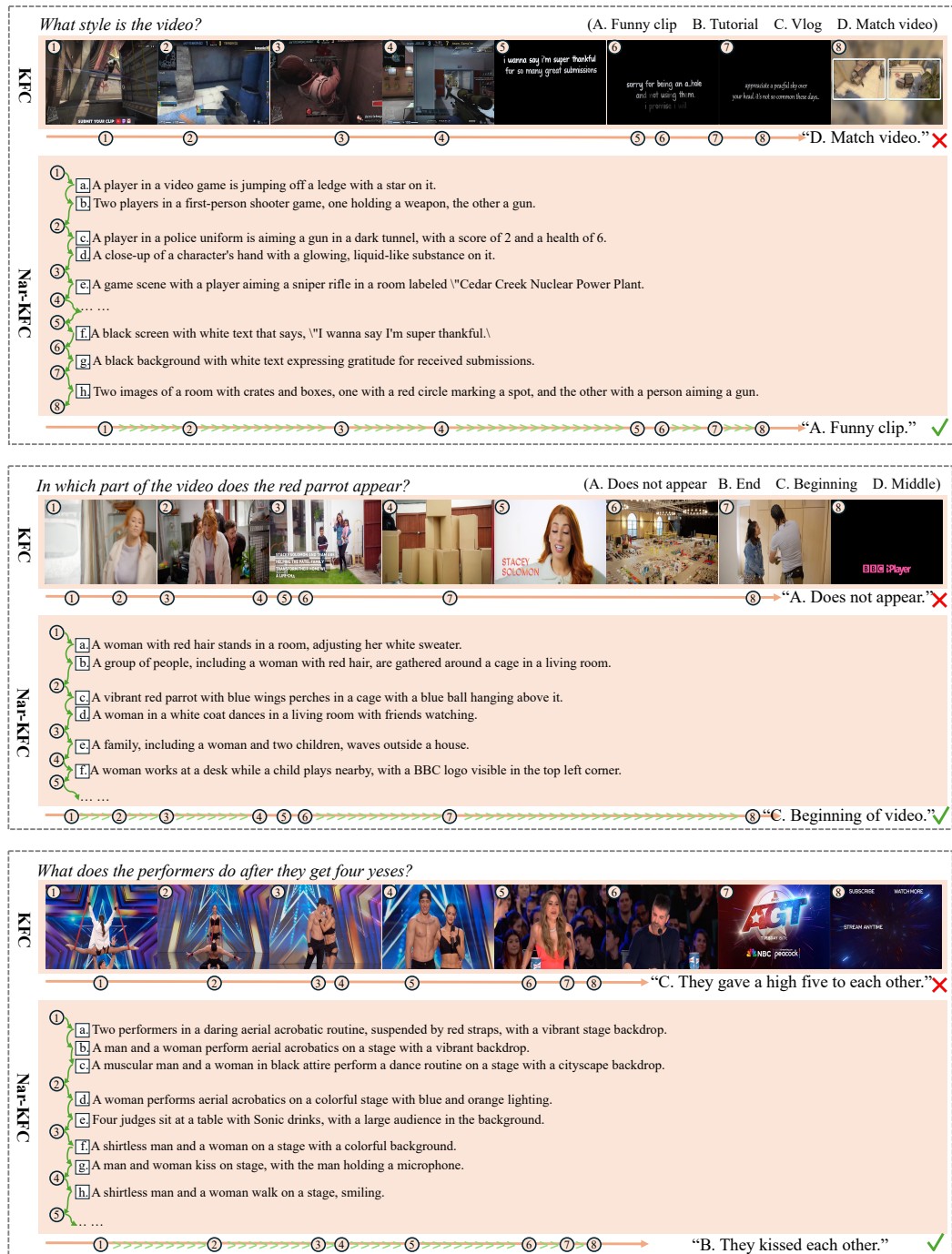

Figure 9: More qualitative examples of our threading keyframe methods Nar-KFC. Zoom in for details.