# OpenReview forum: "Threading Keyframe with Narratives: MLLMs as Strong Long Video Comprehenders"
_ICLR.cc/2026/Conference — ICLR 2026 Poster_

### Official Review · Reviewer_N1Wd · 2025-10-26

**Soundness:** 1
**Presentation:** 3
**Contribution:** 2
**Rating:** 4
**Confidence:** 3

**Summary:**

This paper addresses the challenge of using Multimodal Large Language Models (MLLMs) for long video understanding, where the high number of video frames conflicts with the limited context length of language models. The authors propose Nar-KFC, a plug-and-play module that enhances understanding efficiently. Their method involves two steps: first, selecting keyframes by optimizing for query-relevance and frame-diversity using an efficient greedy search; second, inserting textual narratives from non-key frames to ensure temporal coherence. Nar-KFC acts as a content-aware compression technique, and experiments on various benchmarks show it significantly boosts the performance of existing MLLMs.

**Strengths:**

1. This paper presents a well-written and systematically organized study on a highly practical challenge.
2. The proposed Nar-KFC method offers significant value for real-world deployment as it is a training-free, plug-and-play module that effectively tackles long-video understanding without imposing a substantial computational burden.
3. Nar-KFC's effectiveness is validated through comprehensive experiments on multiple benchmarks, where it demonstrates a notable performance improvement for various MLLMs on tasks like question-answering.

**Weaknesses:**

1. Limited Task Diversity in Evaluation: The proposed method is evaluated exclusively on question-answering (QA) benchmarks. While frame compression may suffice for QA, its efficacy remains unverified for tasks demanding higher visual granularity, such as video captioning or video OCR. Broader evaluation across diverse tasks is needed to fully ascertain the method's applicability and potential limitations.
2. Outdated Baseline Models: The experiments utilize MLLM baselines from the previous year (e.g., Qwen2-VL, InternVL-2). Given the rapid progress in the field, the plug-and-play value of Nar-KFC should be demonstrated on more recent models (e.g., Qwen2.5-VL, InternVL-3) to ensure its contribution remains relevant and complementary to enhanced base capabilities.

**Questions:**

I noticed that all the baselines used in the paper are based on 7B or 8B models. How would the proposed method perform on larger or smaller models, and what impact would model size have on the approach?

---

> ### Author Response · Authors · 2025-11-23
>
> *We thank the reviewer for recognizing the quality and the significance of our work. Major concerns raised in the Weaknesses and Questions are addressed point by point below.*
> ***
>
> ### W1: [**Limited task diversity in evaluation**]
> To address concerns regarding task diversity, we evaluated our methods on additional video captioning benchmarks, including MMbench-Video and MLVU-OpenEnded, beyond standard QA benchmarks. As the proprietary GPT-4 model is required as the judge and the inference is costly, we report results for the two relatively best-performing MLLMs: InternVL3-8B and Qwen3-VL-8B, as also suggested by the reviewer in point 2. All results are based on 8-frame evaluation. Results on the **MMBench-Video** benchmark are presented below.
> |model|CP|FP-S|FP-C|HL|Perception Mean|LR|AR|RR|CSR|TR|Reasoning Mean|Overall|cost|
> |---|:---:|:---:|:---:|:---:|:---:|:---:|:---:|:---:|:---:|:---:|:---:|:---:|:---:|
> |InternVL3-8B|1.78|1.58|1.45|0.29|1.54|1.24|1.98|1.88|1.62|1.42|1.61|1.57|6.8803$|
> |**+ KFC**|1.79|1.64|1.33|0.28|1.56|1.16|1.89|1.85|1.78|1.38|1.58|**1.58**|6.7146$|
> |**+ Nar-KFC**|1.93|1.87|1.58|0.32|1.76|1.41|1.97|1.98|1.86|1.70|1.78|**1.78**|7.4684$|
> |||||||||||||||
> |Qwen3-VL-8B|1.96|1.55|1.43|1.74|1.62|1.17|2.07|1.81|1.69|1.47|1.65|1.64|7.2633$|
> |**+ KFC**|1.95|1.61|1.28|1.79|1.65|1.22|2.14|1.89|1.84|1.62|1.75|**1.69**|7.0038$|
> |**+ Nar-KFC**|1.90|1.74|1.34|2.37|1.75|1.45|1.99|1.93|1.83|1.56|1.76|**1.77**|8.2000$|
>
> GPT-4-1106 is used as the judge model to assess the correspondence between generated outputs and ground truth. For perception, the categories include CP (coarse perception), FP-S (single-instance fine-grained perception), FP-C (cross-instance fine-grained perception), and HL (hallucination). For reasoning, the categories are LR (logic reasoning), AR (attribute reasoning), RR (relation reasoning), CSR (commonsense reasoning), and TR (temporal reasoning).
>
> Additionally, results on **MLVU-OpenEnded** benchmark are presented below:
> |model|sub_scene|summary|G-Avg|cost|
> |---|:---:|:---:|:---:|:---:|
> |InternVL3-8B|5.47|4.40|4.92|10.8059$|
> |**+ KFC**|5.73|4.23|**4.95**|10.0836$|
> |**+ Nar-KFC**|5.69|4.39|**5.02**|11.1204$|
> ||||||
> |Qwen3-VL-8B|6.17|6.01|6.09|10.3736$|
> |**+ KFC**|6.32|5.71|*6.00*|8.4111$|
> |**+ Nar-KFC**|6.28|5.97|**6.12**|10.6716$|
>
> GPT-4-0125  is used as the judge model to evaluate the correspondence between generated outputs and ground truth. On the MLVU-OpenEnded benchmark, KFC-captured keyframes perform worse on the summary task. This is because uniformly sampled frames can cover the entire temporal range of the video, whereas KFC-selected frames may be concentrated within a shorter segment. Nonetheless, our inserted narratives (Nar-KFC) mitigate this limitation by providing supporting and smoothing textual connections among visual tokens.
>
> We also evaluate our methods on extremely short video understanding benchmarks (**TempCompass**, ~10s), where narratives are not required, as well as on the more complex video reasoning benchmark, **Video_Holmes**. The results are shown below.
> |Model|TempC Caption|TempC Overall| <<>>|Video_Holmes|
> |---|:---:|:---:|:---:|:-:|
> |Qwen2.5-VL-7B|74.0|72.2||20.4|
> |**+ KFC**|**74.1**|**72.2**||**20.7**|
> |**+ Nar-KFC**|-|-||**22.9**|
> ||||||
> |InternVL3-8B|80.1|74.8||33.5|
> |**+ KFC**|**80.2**|**74.9**||**34.5**|
> |**+ Nar-KFC**|-|-||**34.5**|
> ||||||
> |Qwen3-VL-8B|79.1|74.4||30.9|
> |**+ KFC**|**80.0**|*74.3*||**33.7**|
> |**+ Nar-KFC**|-|-||**33.8**|
>
> For the relatively short TempCompass benchmark, selecting query-relevant and diversified keyframes results in considerable overlap with uniform sampling, leading to limited performance improvement. In contrast, on the more challenging Video_Holmes benchmark, our approach of carefully selecting keyframes and incorporating threaded narratives significantly enhances the MLLM's video reasoning capabilities.

---

> > ### Author Response · Authors · 2025-11-23
> >
> > ### W2: [**Outdated baseline models**]
> >
> > As suggested by the reviewer, we have updated our evaluations to include more recent VLMs, specifically **Qwen2.5-VL-7B**, **Qwen3-VL-8B,** and **InternVL3-8B**. Notably, the Qwen3-VL-8B model was released on November 1st. Detailed results are presented below.
> > |Model|Short|Medium|Long|VideoMME (no sub / sub)|LVB|MLVU|
> > |---|:---:|:---:|:---:|---|:---:|:---:|
> > |Qwen2.5-VL-7B|65.9 / 66.4|54.4 / 54.3|45.8 / 46.9|55.4 / 55.9|52.7|55.8|
> > |**+ KFC**|68.8 / 70.7|52.6 / 54.9|49.3 / 51.4|**56.9 / 59.0**|**54.3**|**62.6**|
> > |**+ Nar-KFC**|70.1 / 71.0|54.4 / 55.2|49.0 / 49.4|**57.9** / *58.6*|**55.3**|**64.4**|
> > ||||||||
> > |InternVL3-8B|68.7 / 70.9|58.3 / 58.2|50.0 / 50.9|59.0 / 60.0|53.6|60.9|
> > |**+ KFC**|70.9 / 71.9|60.6 / 60.1|50.9 / 51.8|**60.8 / 61.4**|**54.5**|**67.5**|
> > |**+ Nar-KFC**|72.9 / 73.9|62.9 / 62.7|55.7 / 55.8|**63.8 / 64.1**|**54.8**|**68.4**|
> > ||||||||
> > |Qwen3-VL-8B|68.4 / 70.7|55.4 / 55.3|50.1 / 52.0|58.0 / 59.1|54.7|49.5*|
> > |**+ KFC**|68.4/71.9|57.3/57.0|50.8/50.7|**58.9 / 59.9**|**55.8**|**63.0**|
> > |**+ Nar-KFC**|70.4 / 72.9|60.1 / 59.7|52.7 / 52.4|**61.1 / 61.7**|**56.2**|**65.8**|
> >
> > We limit the maximum frame resolution to 256$\times$28$\times$28 during evaluation on the Qwen3-VL-8B model to avoid generating an excessively large number of tokens (e.g., >16,384). **\*** : For the Qwen3-VL baseline result on the MLVU benchmark (i.e., 49.5), we observed that the model frequently refuses to select an answer from the provided candidates, often outputting statements such as “None of the options can answer the question since the provided video frames are not sufficient.” This behavior resulted in many answers with incorrect formats and significantly lower performance. In contrast, our KFC-selected keyframes effectively address this issue by presenting more relevant video content.
> >
> > We are editing our next version paper and updating main results with newer MLLMs.
> >
> > ***
> > ### Q1: [**Scaling to larger size models**]
> > We extend our Nar-KFC framework to two larger models: LLaVA-OneVision-72B (32 frames) and LLaVA-Video-72BQwen2 (64 frames). We compare their performance with state-of-the-art proprietary models and VideoLLMs trained with full-parameter fine-tuning. The results on the Video-MME benchmark are presented below.
> > |Model|Frames|Short|Medium|Long|Video-MME (no sub)|
> > |---|:---:|:---:|:---:|:---:|---|
> > |VideoChat-Flash@448-7B [1]|N/A|-|-|-|65.3|
> > |LLaVA-OneVision-72B + T* [2]|32|77.5|66.6|61.0|68.3|
> > |ViLAMP-7B [3]|1 fps|-|-|-|67.5|
> > |Aria-8x3.5B|256|76.9|67.0|58.8|67.6|
> > |GPT-4o (0615)|384|80.0|70.3|65.3|71.9|
> > |AdaReTake-72B [4]|2fps|-|-|-|73.5|
> > |Gemini-1.5-Pro (0615)|1/0.5 fps|81.7|74.3|67.4|75.0|
> > |||||||
> > |LLaVA-OneVision-72B|32|76.7|62.2|60.0|66.3|
> > |**+ Nar-KFC**|32|**77.5**|**68.6**|**61.9**|**69.6**|
> > |LLaVA-Video-72B|64|81.7|67.9|61.8|70.4|
> > |**+ Nar-KFC**|64|**82.0**|**68.9**|**63.6**|**71.5**|
> >
> > Meanwhile, performance on the MLVU benchmark with 72B models is shown below:
> > |Model|Frames|MLVU|
> > |---|:---:|---|
> > |GPT-4o|0.5fps|64.6|
> > |VideoLLaMA3-7B|<=180|73.0|
> > |VideoChat-Flash@448-7B [1]|1fps|74.7|
> > |AdaReTake-72B [4]|2fps|78.1|
> > |||
> > |LLaVA-OneVision-72B|32|66.4|
> > |**+Nar-KFC**|32|**74.4**|
> > |LLaVA-Video-72B|64|74.4(73.6)*|
> > |**+Nar-KFC**|64|**75.0**|
> >
> > **\***: 74.4 is reported in huggingface while 73.6 is obtained from our own implementation.
> > As observed, despite utilizing significantly fewer frames, our Nar-KFC generally achieves competitive, on-par performance compared to full-parameter tuned VideoLLMs trained on thousands of frames, and proprietary models such as GPT-4o. We have incorporated these larger model results in Appendix D.3.
> >
> > [1] VideoChat-Flash: Hierarchical compression for long-context video modeling. arXiv 2024.
> > [2] Re-thinking temporal search for long-form video understanding. CVPR 2025.
> > [3] Scaling video-language models to 10k frames via hierarchical differential distillation. ICML 2025
> > [4] AdaReTake: Adaptive redundancy reduction to perceive longer for video-language understanding. arXiv 2025.

---

> > > ### Comment · Reviewer_N1Wd · 2025-11-24
> > > **Response to Authors' Rebuttal**
> > >
> > > Thank you for your detailed response and clarifications. You have addressed some of my concerns. However, one remaining point is regarding the diversity of tasks. I feel that the presented results still lack evaluation on benchmarks such as video captioning, which demands fine-grained detail understanding. If you could provide equally convincing results on Dream1K, I would be willing to increase my score.

---

> > > > ### Author Response · Authors · 2025-11-26
> > > >
> > > > We sincerely thank the reviewer for their prompt feedback. As suggested, we have supplemented our work with additional experiments on the video captioning benchmark Dream1K. Since the mainstream MLLM inference frameworks (VLMEvalKit and LMMs-Eval) do not include Dream1K, we utilized the Tarsier2 repository (https://github.com/bytedance/tarsier/tree/main) for evaluation. We first re-implemented the evaluation process based on the Tarsier2-7B model. Notably, the default judge model (GPT-3.5-turbo) has been deprecated and updated to GPT-4.1-mini (2025-04-14), resulting in bit differences between our reproduced results and those originally reported in the repository. Detailed results for the baseline Tarsier2-7B model are presented below.
> > > >
> > > > | Task                    | F1 Score | Action Recall | Action Precision | Success | Failed |
> > > > |:-------------------|----------|---------------|------------------|---------|--------|
> > > > | DREAM/movie_animation   | 0.28     | 0.323         | 0.247            | 190     | 10     |
> > > > | DREAM/movie_live_action | 0.326    | 0.372         | 0.29             | 197     | 3      |
> > > > | DREAM/vertical_video    | 0.32     | 0.301         | 0.341            | 196     | 4      |
> > > > | DREAM/videvo            | 0.355    | 0.372         | 0.34             | 195     | 5      |
> > > > | DREAM/youtube           | 0.258    | 0.263         | 0.253            | 196     | 4      |
> > > > | OVERALL                 | 0.309    | 0.326         | 0.294            | 974     | 26     |
> > > >
> > > > Building on these reproduced baselines, we evaluated the effectiveness of our approach on the two best-performing models, Tarsier2-7B and Tarsier2-ReCap-7B. Each video was captioned at 5 fps to ensure sufficient narrative detail. The results are reported below.
> > > > | Model                | F1 Score | Action Recall | Action Precision |
> > > > |----------------------|----------|---------------|------------------|
> > > > | Tarsier2-ReCap-7B    | 30.1     | 31.7          | 28.6         |
> > > > |  **+ Nar-KFC**       | **31.5**     | **35.4**         | *28.4*            |
> > > > | Tarsier2-7B-0115     | 30.9     | 32.6          | 29.4          |
> > > > | **+ Nar-KFC**          | **32.9**     | **37.1**      | **29.6**        |
> > > >
> > > > **Observations and analysis:**
> > > >
> > > > Our keyframe selection method (KFC) yielded results nearly identical to uniform sampling (not shown in the table). This can be attributed to two main factors: (1) The average video length in Dream1K is only 7.3 seconds, resulting in significant overlap between the frames selected by KFC and uniform sampling. (2) During evaluation, all videos were tested using the same query-agnostic prompt (“Describe the video in detail”), which prevented our query-relevant score from being utilized. As a result, selection based solely on diversity also produced frame sets similar to the uniform setting. However, we observe consistent improvements when incorporating narratives (+Nar-KFC), which captures more fine-grained information and achieves effects comparable to increasing the number of frames.
> > > >
> > > > Finally, we want to respectfully remind the reviewer that *Dream1K looks too short (7.3s)* to truly demonstrate the advantage of our methods, as KFC and Nar-KFC are proposed primarily for *long videos (at minute or hour scale)*. Additionally, we would like to kindly note that our initial response included results **on two captioning-style benchmarks, MMBench-Video and MLVU-OpenEnded**. Again, we sincerely appreciate the reviewer’s valuable feedback and constructive suggestions.

---

> > > > > ### Comment · Reviewer_N1Wd · 2025-11-27
> > > > > **Response to New Rebuttal**
> > > > >
> > > > > Thank you for the additional experimental results! According to all the information you have provided, I would like to raise my rating to 6. I also hope these results will be added to the final version.

---

> > > > > > ### Author Response · Authors · 2025-11-27
> > > > > >
> > > > > > Thank you very much for your appreciation of our work! We are currently revising our paper to update the main results with newer MLLMs and to incorporate additional benchmarks, as you suggested. We sincerely appreciate your valuable feedback and suggestions.

---

### Official Review · Reviewer_co9R · 2025-11-01

**Soundness:** 3
**Presentation:** 3
**Contribution:** 3
**Rating:** 6
**Confidence:** 4

**Summary:**

This paper introduces Narrating KeyFrames Capturing (Nar-KFC), a plug-and-play module designed to enhance long video understanding in multimodal large language models (MLLMs). Nar-KFC addresses the keyframe selection problem by formulating it as an integer linear programming task that jointly optimizes query relevance and frame diversity. To reduce computational cost, it also offers a greedy search alternative. Additionally, Nar-KFC leverages an off-the-shelf captioning MLLM to generate descriptive captions for the keyframes. A simplified variant, KFC, which omits the captioner, is also evaluated for ablation. Both Nar-KFC and KFC are tested across multiple backbone MLLMs and benchmark datasets, demonstrating their effectiveness and versatility.

**Strengths:**

* The paper is clearly written and well-structured.
* Nar-KFC demonstrates consistent performance gains across different MLLMs, effectively validating its utility.
* Exploring improved frame sampling strategies for long video understanding is a valuable research direction.

**Weaknesses:**

* Unlike token compression approaches, Nar-KFC relies primarily on frame selection for information compression. The optimization criteria, i.e. frame-level diversity and query-frame relevance, are computed at the global frame level. This raises concerns about scenarios involving subtle or localized changes (e.g., small objects evolving over time), where high overall frame similarity might cause truly informative frames to be inadvertently discarded, as such nuances may not be captured by coarse frame-level metrics.
* The most significant performance improvements appear to stem from the captioner module, which enriches keyframes with descriptive text. However, this component is computationally expensive, potentially offsetting the efficiency gains expected from reduced frame counts.
* Since the method does not incorporate any token-level reduction mechanism, the number of selected keyframes is inherently constrained by the MLLM’s effective context length and its capacity to handle multiple images, shaped by its pre-training and SFT. This limitation could become problematic for extremely long videos with frequent scene transitions, where even a moderate number of keyframes may exceed effective modeling capacity. A more comprehensive comparison with token-reduction-based methods (rather than only keyframe selection approaches) would better contextualize Nar-KFC’s trade-offs between performance, robustness, and scalability.

**Questions:**

* Using Qwen2-VL as the MLLM and captioner at the same time leads to performance degradation, I wonder if the authors can provide more analysis on that matter.

---

> ### Author Response · Authors · 2025-11-23
>
> *We thank the reviewer for recognizing the quality of our paper. Major concerns raised in the Weaknesses and Questions are addressed point by point below.*
> ***
> ### W1: [**Risk of discarding subtle or localized changes**]
>
> We understand the reviewer’s concern regarding current keyframe selection methods, where globally sampled frames may overlook subtle or localized changes occurring between frames. To quantitatively assess this issue, we conducted additional experiments on a **video captioning task (MMBench-Video) that requires fine-grained understanding and reasoning**. GPT-4-0125 is used as the judge model to evaluate answer quality.
> |model|CP|FP-S|FP-C|HL|Perception Mean|LR|AR|RR|CSR|TR|Reasoning Mean|Overall|cost|
> |---|:---:|:---:|:---:|:---:|:---:|:---:|:---:|:---:|:---:|:---:|:---:|:---:|:---:|
> |InternVL3-8B|1.78|1.58|1.45|0.29|1.54|1.24|1.98|1.88|1.62|1.42|1.61|1.57|6.8803$|
> |**+ KFC**|1.79|1.64|1.33|0.28|1.56|1.16|1.89|1.85|1.78|1.38|1.58|**1.58**|6.7146$|
> |**+ Nar-KFC**|1.93|1.87|1.58|0.32|1.76|1.41|1.97|1.98|1.86|1.70|1.78|**1.78**|7.4684$|
> |||||||||||||||
> |Qwen3-VL-8B|1.96|1.55|1.43|1.74|1.62|1.17|2.07|1.81|1.69|1.47|1.65|1.64|7.2633$|
> |**+ KFC**|1.95|1.61|1.28|1.79|1.65|1.22|2.14|1.89|1.84|1.62|1.75|**1.69**|7.0038$|
> |**+ Nar-KFC**|1.90|1.74|1.34|2.37|1.75|1.45|1.99|1.93|1.83|1.56|1.76|**1.77**|8.2000$|
>
> For perception, the categories include CP (coarse perception), FP-S (single-instance fine-grained perception), FP-C (cross-instance fine-grained perception), and HL (hallucination). For reasoning, the categories are LR (logic reasoning), AR (attribute reasoning), RR (relation reasoning), CSR (commonsense reasoning), and TR (temporal reasoning).
>
> We focus primarily on two fine-grained perception settings: **FP-S** and **FP-C**. *Notably, our KFC and Nar-KFC methods significantly improve FP-S performance, as the selected keyframes help accurately localize specific single instances.* However, for *FP-C, where multiple video event instances are required, KFC does not perform as well, suggesting that our frame selection may miss subtle or detailed information across instances*. Despite this limitation, both KFC and Nar-KFC consistently show robust improvements on fine-grained video captioning tasks.
>
> In conclusion, we acknowledge the potential limitation of discarding fine-grained information due to global frame selection. However, we believe that capturing **long-term dependencies and correlations **is more critical for comprehensive video understanding. Our consistent improvements on video captioning tasks demonstrate that the loss of fine-grained information does not severely impact overall effectiveness.
> ***
>
> ### W2: [**Computationally expensive captioner module**]
>
> We present a detailed analysis of the computational complexity (TFLOPs, latency, and memory usage) for extracting CLIP embeddings and frame narratives, as shown in the table below. The results are based on an average 17-minute video (1,020 frames at 1 fps) in Video-MME.
> |Model|Frames|TFLOPs↓|Latency(s)↓|Memory(GB)|
> |---|:---:|:---:|:---:|:---:|
> |*Offline Extraction*|||||
> |CLIP-ViT-L-336px|1020|420.8|25.8|1.6|
> |Qwen2-VL-2B|1020|4462.5|360.5|7.2|
> |*On-demand Video System*|||||
> |Qwen2-VL-2B|≤210|≤918.8|≤74.2|7.2|
>
> We analyze the overall computational complexity of the captioner module from two perspectives: **offline** caption extraction and **on-demand video system** processing.
>
> **A**. Since our captioning process is performed offline before MLLM video inference, the high computational cost (e.g., 4462.5 TFLOPs) does not impact inference speed. Additionally, our inserted narratives contain far fewer tokens compared to visual frames, as shown in Table 5. Therefore, the additional inference cost is acceptable given the long-range video content these narratives cover. The introduced textual cues offer an effective solution for understanding long videos with limited contextual length.
>
> **B**. For on-demand (long) video understanding, our lightweight captioner only needs to extract at most 210 narratives, regardless of video length (as demonstrated in our paper, adding more narrations does not yield further improvements and may exceed the context length of MLLMs). The caption extraction process requires less than 74.2 sec of latency. In practice, there are often no more than 210 frames between the first and last sampled keyframes, which can further reduce preprocessing time. The low computational cost of captioning is primarily due to our lightweight captioner, as we demonstrate that Nar-KFC’s performance is not sensitive to captioner size and only a small number of frames are processed.

---

> > ### Author Response · Authors · 2025-11-23
> >
> > ### W3: [**Comparisons with token-reduction-based methods**]
> >
> > As suggested by the reviewer, we compare our methods with those **token-reduction-based approaches** (e.g., VideoChat-Flash[1], T*[2], ViLAMP[3], AdaReTake[4] ) and advanced proprietary models (e.g., GPT-4o, Gemini-1.5-pro) on the Video-MME and MLVU benchmarks. Results on the Video-MME benchmark are shown below.
> > |Model|Frames|Short|Medium|Long|Video-MME (no sub)|
> > |---|:---:|:---:|:---:|:---:|---|
> > |VideoChat-Flash@448-7B [1]|N/A|-|-|-|65.3|
> > |LLaVA-OneVision-72B + T* [2]|32|77.5|66.6|61.0|68.3|
> > |ViLAMP-7B [3]|1 fps|-|-|-|67.5|
> > |Aria-8x3.5B|256|76.9|67.0|58.8|67.6|
> > |GPT-4o (0615)|384|80.0|70.3|65.3|71.9|
> > |AdaReTake-72B [4]|2fps|-|-|-|73.5|
> > |Gemini-1.5-Pro (0615)|1/0.5 fps|81.7|74.3|67.4|75.0|
> > |||||||
> > |LLaVA-OneVision-72B|32|76.7|62.2|60.0|66.3|
> > |**+ Nar-KFC**|32|**77.5**|**68.6**|**61.9**|**69.6**|
> > |LLaVA-Video-72B|64|81.7|67.9|61.8|70.4|
> > |**+ Nar-KFC**|64|**82.0**|**68.9**|**63.6**|**71.5**|
> >
> > Meanwhile, performance on the MLVU benchmark with 72B models is shown below:
> > |Model|Frames|MLVU|
> > |---|:---:|---|
> > |GPT-4o|0.5fps|64.6|
> > |VideoLLaMA3-7B|<=180|73.0|
> > |VideoChat-Flash@448-7B [1]|1fps|74.7|
> > |AdaReTake-72B [4]|2fps|78.1|
> > |||
> > |LLaVA-OneVision-72B|32|66.4|
> > |**+Nar-KFC**|32|**74.4**|
> > |LLaVA-Video-72B|64|74.4(73.6)*|
> > |**+Nar-KFC**|64|**75.0**|
> >
> > **\***: 74.4 is reported in huggingface while 73.6 is obtained from our own implementation. It is important to see that token-reduction-based methods typically require full-parameter tuning of large MLLMs on thousands of frames, which is particularly time-consuming. For example, ViLAMP requires 32 A100 GPUs and two weeks of training for just one epoch.
> >
> > Despite using significantly fewer frames (32 or 64 frames compared to thousands), our Nar-KFC method achieves competitive and comparable performance with these fully fine-tuned VideoLLMs and proprietary models. This demonstrates the robustness and scalability of our approach. These comparative results have been included in Appendix D.2.
> >
> > [1] VideoChat-Flash: Hierarchical compression for long-context video modeling. arXiv 2024.
> > [2] Re-thinking temporal search for long-form video understanding. CVPR 2025.
> > [3] Scaling video-language models to 10k frames via hierarchical differential distillation. ICML 2025.
> > [4] AdaReTake: Adaptive redundancy reduction to perceive longer for video-language understanding. arXiv 2025.
> >
> >
> > ***
> > ### Q1: [**Performance degradation with Qwen2-VL**]
> >
> > Using Qwen2-VL as both the captioner and inference model may result in **overlapping or homogeneous content** between keyframes and narratives. In contrast, employing different MLLMs for captioning and inference can supplement information and knowledge from diverse sources. Our experiments show that narratives generated by Qwen2-VL-2B can also enhance the performance of Qwen2.5-VL and Qwen3-VL (please refer to our response 2 to reviewer A1NG), which supports this analysis.
> >
> > Additionally, in the Qwen2-VL architecture, the **maximum image resolution** is manually set, often resulting in a significantly larger number of frame patch tokens (for enhancing performance) compared to text tokens (narratives). We conjecture that this imbalance may reduce the impact of the inserted narratives (Nar-KFC) on overall performance

---

> > > ### Author Response · Authors · 2025-11-27
> > >
> > > Dear reviewer co9R,
> > >
> > > I hope this message finds you well. As the discussion period is nearing its end, I wanted to ensure we have addressed all your concerns satisfactorily. If there are any additional points or feedback you'd like us to consider, please let us know. Your insights are invaluable to us, and we're eager to address any remaining issues to improve our work.
> > >
> > > Thank you for your time and effort in reviewing our paper.

---

### Official Review · Reviewer_TdJ3 · 2025-11-01

**Soundness:** 3
**Presentation:** 3
**Contribution:** 3
**Rating:** 4
**Confidence:** 3

**Summary:**

This paper introduces Nar-KFC, a novel and practical training-free framework designed to enhance long-video comprehension for existing Multimodal Large Language Models (MLLMs) constrained by limited context windows.

**Strengths:**

1.This paper presents Nar-KFC, a hybrid representation method that interleaves visual keyframes with textual narratives, offering a novel perspective on video compression and long-video understanding. The authors formulate keyframe selection as an Integer Quadratic Programming (IQP) problem, systematically optimizing both query relevance and frame diversity, making it a more principled alternative to heuristic approaches.

2.The proposed method holds strong practical value: it is a plug-and-play, training-free module that can be directly applied to existing multimodal large language models (MLLMs) to significantly enhance long-video comprehension without retraining, opening a new direction for efficient hybrid video representation.

**Weaknesses:**

1. A significant potential risk of this method is that the lightweight captioner used for narrative generation may introduce errors or hallucinations. More importantly, this captioner operates in a "query-agnostic" manner; it merely describes the content of non-keyframe segments, and this content may be entirely irrelevant to the user's specific query. This results in the Nar-KFC method actively injecting a substantial volume of irrelevant noise into the MLLM's context . If a given narrative happens to contradict the visual content of the keyframes or the query itself, this could severely mislead the MLLM.

2. The keyframe selection is entirely dependent on the quality of the upstream VLM (CLIP) used to compute the $S_{QR}$ and $S_{FD}$ scores. This reliance is potentially unreliable, as the model may fail to select rational keyframes for queries involving complex relations or fine-grained details. Furthermore, the MLLM models employed for the experiments (e.g., InternVL2, Qwen2-VL) appear to be outdated.

**Questions:**

1. The query-agnostic captioner may poses risks of noise and hallucination. Could the authors provide a failure analysis quantifying how often MLLM errors stem from misleading narratives?

2. The keyframe selection relies entirely on CLIP, which may be unreliable for complex or fine-grained queries, and the MLLM baselines used are outdated. Could the authors adopt a more advanced VLM for selection and validate Nar-KFC on a recent MLLM (e.g., Qwen2.5-VL)? It would also be valuable to show performance differences when using newer VLMs, highlighting how the framework scales with stronger vision-language backbones.

---

> ### Author Response · Authors · 2025-11-23
>
> *We thank the reviewer for recognizing the novelty and the significance of our work. Major concerns raised in the Weaknesses and Questions are addressed point by point below.*
>
> ***
> ### W1 & Q1: [**Risk of injecting irrelevant noise from query-agnostic narratives**]
>
> We appreciate the reviewer’s comment that “the captioner operates in a query-agnostic manner, which might inject irrelevant noise into the MLLM’s context.” We agree with this concern, as long videos may be associated with multiple potential questions, and extracting query-relevant narratives for each question would be complex and computationally expensive, even with lightweight captioners.
>
> However, our results demonstrate that keyframes play a dominant role in long video understanding, while narratives primarily serve as auxiliary context to support keyframe comprehension (see L458). The presence of potentially irrelevant narratives does not substantially occupy the MLLM’s context, as the number of text tokens is significantly lower than that of frame patch tokens (see Table 5).
>
> To **quantitatively assess** the risk of injecting query-irrelevant or contradictory narratives, we compare performance across detailed task types from multiple benchmarks, evaluating settings **without narratives (KFC)** and **with narratives (Nar-KFC)**.
> |Video task type|Uniform|KFC|Nar-KFC|Drop?|
> |---|:---:|:---:|:---:|:---:|
> |Temporal Perception|52.7|61.8|63.6||
> |Spatial Perception|61.1|59.3|57.4|Y|
> |Attribute Perception|63.5|69.8|72.1||
> |Action Recognition|47.0|47.6|48.6||
> |Object Recognition|56.8|59.3|66.1||
> |OCR Problems|56.8|59.0|70.5||
> |Count Problems|35.4|32.1|41.8||
> |Temporal Reasoning|38.4|37.3|37.9|Y|
> |Spatial Reasoning|69.6|73.2|73.2|-|
> |Action Reasoning|46.0|48.8|49.5||
> |Object Reasoning|51.1|54.2|54.8||
> |Information Synopsis|68.7|74.0|75.2||
> ||||||
> |plotQA|56.8|58.8|66.2||
> |needle|50.7|78.3|78.9||
> |ego|50.9|55.4|55.0|Y|
> |count|35.9|46.1|46.7||
> |order|42.1|52.5|53.3||
> |anomaly|64.0|65.0|66.0||
> |topic|77.9|76.0|83.3||
>
> We use a “Y” label to indicate any performance decrease that may suggest a potential failure introduced by Nar-KFC narratives compared to Uniform and KFC settings. Among 19 video-related tasks evaluated, such failures were observed in only 3 cases, with negligible negative impact.
> Therefore, we conclude that the likelihood of inserted contradictory narratives misleading MLLMs is low, and the overall positive effect of including narratives outweighs the potential negative impact.
>
> ***
> ### W2 & Q2 (a): [**Potentially unreliable frame selection via CLIP**]
> As suggested by the reviewer, we present results using different (and intuitively more powerful) VLMs for top-K keyframe selection. The comparisons are shown below.
> |Model|Video-MME(no sub/sub)|
> |---|:---:|
> |CLIP-ViT-L-336px|47.7/50.0|
> |SigLIP-so400m-patch14-384|47.3/51.0|
> |BLIP-2|47.8/50.9|
>
> As observed,  more advanced VLMs such as BLIP-2 may provide slightly more accurate frames, but the overall differences among various VLM selectors are minimal and can be considered negligible. This is because keyframe selection primarily serves to localize relevant frames or regions, whereas the understanding of complex relations or fine-grained details relies mainly on the capabilities of the inference MLLM. Given the high similarity among adjacent video frames, it is generally sufficient for VLM selectors to provide globally relevant and near-correct keyframes.
>
> Additionally, CLIP (or SigLIP, etc.) is commonly used for long video keyframe selection in previous literature due to its lightweight architecture and fast inference. For long videos containing thousands of frames, extracting frame-level or token-level embeddings using larger VLMs (with billions of parameters) would be extremely time-consuming.
> |Method|Venue|VLM selector choice|
> |---|:---:|:---:|
> |BOLT[1]|CVPR2025|CLIP-L/14|
> |AKS[2]|CVPR2025|BLIP / CLIP|
> |MDP3[3]|ICCV2025|SigLIP|
> |Flow4Agent[4]|ICCV2025|SigLIP|
>
> To further address the reviewer’s concern regarding our method’s ability to handle **complex relations** and **fine-grained details**, we have supplemented our results with additional evaluations on video captioning datasets (which require fine-grained understanding) and a video reasoning benchmark (which requires reasoning over complex relations). We kindly refer the reviewer to our response 1 for reviewer N1Wd for detailed results and analysis.
>
> [1] BOLT: Boost Large Vision-Language Model Without Training for Long-form Video Understanding. CVPR 2025.
> [2] Adaptive Keyframe Sampling for Long Video Understanding. CVPR 2025.
> [3] MDP3: A Training-free Approach for List-wise Frame Selection in Video-LLMs. ICCV 2025.
> [4] Flow4Agent: Long-form Video Understanding via Motion Prior from Optical Flow. ICCV 2025.

---

> > ### Author Response · Authors · 2025-11-23
> >
> > ### W2 & Q2 (b): [**Validate Nar-KFC on more recent MLLMs (e.g., Qwen2.5-VL)**]
> > As suggested by the reviewer, we conduct further evaluations on more recent VLMs including **Qwen2.5-VL-7B**, **Qwen3-VL-8B**, and **InternVL3-8B**. Notably, the Qwen3-VL-8B model was released on November 1st. Detailed results are presented below.
> > |Model|Short|Medium|Long|VideoMME (no sub / sub)|LVB|MLVU|
> > |---|:---:|:---:|:---:|---|:---:|:---:|
> > |Qwen2.5-VL-7B|65.9 / 66.4|54.4 / 54.3|45.8 / 46.9|55.4 / 55.9|52.7|55.8|
> > |**+ KFC**|68.8 / 70.7|52.6 / 54.9|49.3 / 51.4|**56.9 / 59.0**|**54.3**|**62.6**|
> > |**+ Nar-KFC**|70.1 / 71.0|54.4 / 55.2|49.0 / 49.4|**57.9** / *58.6*|**55.3**|**64.4**|
> > ||||||||
> > |InternVL3-8B|68.7 / 70.9|58.3 / 58.2|50.0 / 50.9|59.0 / 60.0|53.6|60.9|
> > |**+ KFC**|70.9 / 71.9|60.6 / 60.1|50.9 / 51.8|**60.8 / 61.4**|**54.5**|**67.5**|
> > |**+ Nar-KFC**|72.9 / 73.9|62.9 / 62.7|55.7 / 55.8|**63.8 / 64.1**|**54.8**|**68.4**|
> > ||||||||
> > |Qwen3-VL-8B|68.4 / 70.7|55.4 / 55.3|50.1 / 52.0|58.0 / 59.1|54.7|49.5*|
> > |**+ KFC**|68.4/71.9|57.3/57.0|50.8/50.7|**58.9 / 59.9**|**55.8**|**63.0**|
> > |**+ Nar-KFC**|70.4 / 72.9|60.1 / 59.7|52.7 / 52.4|**61.1 / 61.7**|**56.2**|**65.8**|
> >
> > We limit the maximum frame resolution to 256$\times$28$\times$28 during evaluation on the Qwen3-VL-8B model to avoid generating an excessively large number of tokens (e.g., >16,384). **\*** : For the Qwen3-VL baseline result on the MLVU benchmark (i.e., 49.5), we found that the model often refuses to select an option from the given candidates, outputting “None of the options can answer the question since the provided video frames are not sufficient.” This leads to many incorrect answers and significantly lower performance. In contrast, our KFC-selected keyframes effectively address this issue by providing query-relevant video content.
> >
> > We are editing our next version paper and updating main results with newer MLLMs.

---

> > > ### Author Response · Authors · 2025-11-27
> > >
> > > Dear reviewer TdJ3,
> > >
> > > I hope this message finds you well. As the discussion period is nearing its end, I wanted to ensure we have addressed all your concerns satisfactorily. If there are any additional points or feedback you'd like us to consider, please let us know. Your insights are invaluable to us, and we're eager to address any remaining issues to improve our work.
> > >
> > > Thank you for your time and effort in reviewing our paper.

---

### Official Review · Reviewer_A1NG · 2025-11-01

**Soundness:** 3
**Presentation:** 3
**Contribution:** 2
**Rating:** 6
**Confidence:** 4

**Summary:**

This paper introduces Nar-KFC, a plug-and-play module designed to improve long video understanding for Multimodal Large Language Models (MLLMs). Nar-KFC selects representative keyframes by jointly optimizing relevance and diversity through an efficient greedy search, then enriches temporal coherence by inserting textual narratives from non-keyframes. This dual strategy achieves both temporal consistency and content compression, enabling MLLMs to process long videos more effectively. Extensive experiments on multiple benchmarks show that Nar-KFC significantly boosts MLLM performance with minimal computational overhead.

**Strengths:**

1. Well-written and easy to follow.

2. Keyframe selection is formulated as an integer quadratic programming problem with a customized greedy search, providing clear theoretical grounding rather than relying on heuristic rules.

3. The plug-and-play design is flexible and compatible with various MLLMs, training-free, which reduces computation cost and overfitting risk.

4. Demonstrates consistent performance gains across different models (e.g., InternVL2, Qwen2-VL) and model sizes, showing its generalizability.

**Weaknesses:**

My main concerns are with the experiments:

1. The number of benchmarks is limited. As a training-free module, Nar-KFC needs more diverse benchmarks to convincingly demonstrate its generality and robustness.

2. The base models are outdated — the strongest one, LLaVA-Video, is already a year old. Including 1–2 more recent VLMs (e.g., Qwen2.5-VL, Qwen3-VL, Intern3-VL) would strengthen the claims.

3. Some baselines in Table 1 seem questionable; for instance, LLaVA-Video’s VideoMME performance should be higher than 55.9 / 56.7.

4. The paper should discuss and compare with more recent keyframe selection methods [1][2] to contextualize the contributions.

[1] Flow4Agent: Long-form Video Understanding via Motion Prior from Optical Flow. ICCV25

[2] From Trial to Triumph: Advancing Long Video Understanding via Visual Context Sample Scaling and Self-reward Alignment. ICCV25

**Questions:**

Please refer to the Weaknesses section. The authors should seriously address the concerns, especially the experimental aspects, as failing to do so may negatively impact the paper’s evaluation.

---

> ### Author Response · Authors · 2025-11-23
>
> *We thank the reviewer for recognizing the quality and the significance of our work. Major concerns raised in the Weaknesses and Questions are addressed point by point below.*
>
> ***
> ### W1: [**Results on more benchmarks**]
> Beyond standard QA benchmarks, we further evaluate our methods on additional video captioning benchmarks, including **MMbench-Video** and **MLVU-OpenEnded**. Since the proprietary GPT-4 model is required as the judge model and the evaluation process is costly, we report results on two leading MLLMs: InternVL3-8B and Qwen3-VL-8B, as also suggested by the reviewer in point 2. All results are based on 8-frame evaluation. Results on **MMBench-Video** are presented below.
> |model|CP|FP-S|FP-C|HL|Perception Mean|LR|AR|RR|CSR|TR|Reasoning Mean|Overall|cost|
> |---|:---:|:---:|:---:|:---:|:---:|:---:|:---:|:---:|:---:|:---:|:---:|:---:|:---:|
> |InternVL3-8B|1.78|1.58|1.45|0.29|1.54|1.24|1.98|1.88|1.62|1.42|1.61|1.57|6.8803$|
> |**+ KFC**|1.79|1.64|1.33|0.28|1.56|1.16|1.89|1.85|1.78|1.38|1.58|**1.58**|6.7146$|
> |**+ Nar-KFC**|1.93|1.87|1.58|0.32|1.76|1.41|1.97|1.98|1.86|1.70|1.78|**1.78**|7.4684$|
> |||||||||||||||
> |Qwen3-VL-8B|1.96|1.55|1.43|1.74|1.62|1.17|2.07|1.81|1.69|1.47|1.65|1.64|7.2633$|
> |**+ KFC**|1.95|1.61|1.28|1.79|1.65|1.22|2.14|1.89|1.84|1.62|1.75|**1.69**|7.0038$|
> |**+ Nar-KFC**|1.90|1.74|1.34|2.37|1.75|1.45|1.99|1.93|1.83|1.56|1.76|**1.77**|8.2000$|
>
> GPT-4-1106 is used as the judge model to evaluate the correspondence between generations and ground truth. For perception categories: CP (coarse perception), FP-S (single-instance fine-grained perception), FP-C (cross-instance fine-grained perception), HL (hallucination). For reasoning categories: LR (logic reasoning), AR (attribute reasoning), RR (relation reasoning), CSR (commonsense reasoning), TR (temporal reasoning).
>
> Additionally, results on the **MLVU-OpenEnded** benchmark are shown below:
> |model|sub_scene|summary|G-Avg|cost|
> |---|:---:|:---:|:---:|:---:|
> |InternVL3-8B|5.47|4.40|4.92|10.8059$|
> |**+ KFC**|5.73|4.23|**4.95**|10.0836$|
> |**+ Nar-KFC**|5.69|4.39|**5.02**|11.1204$|
> ||||||
> |Qwen3-VL-8B|6.17|6.01|6.09|10.3736$|
> |**+ KFC**|6.32|5.71|*6.00*|8.4111$|
> |**+ Nar-KFC**|6.28|5.97|**6.12**|10.6716$|
>
> GPT-4-0125  is used as the judge model to evaluate the correspondence between generated outputs and ground truth. On the MLVU-OpenEnded benchmark, KFC-captured keyframes perform worse on the summary task. This is because uniformly sampled frames can cover the entire temporal range of the video, whereas KFC-selected frames may be concentrated within a shorter segment. Nonetheless, our inserted narratives (Nar-KFC) mitigate this limitation by providing supporting and smoothing textual connections among visual tokens.
>
> We also evaluate our methods on extremely short video understanding benchmarks (**TempCompass**, ~10s), where narratives are not required, as well as on the more complex video reasoning benchmark, **Video_Holmes**. The results are shown below.
> |Model|TempC Caption|TempC Overall| <<>>|Video_Holmes|
> |---|:---:|:---:|:---:|:-:|
> |Qwen2.5-VL-7B|74.0|72.2||20.4|
> |**+ KFC**|**74.1**|**72.2**||**20.7**|
> |**+ Nar-KFC**|-|-||**22.9**|
> ||||||
> |InternVL3-8B|80.1|74.8||33.5|
> |**+ KFC**|**80.2**|**74.9**||**34.5**|
> |**+ Nar-KFC**|-|-||**34.5**|
> ||||||
> |Qwen3-VL-8B|79.1|74.4||30.9|
> |**+ KFC**|**80.0**|*74.3*||**33.7**|
> |**+ Nar-KFC**|-|-||**33.8**|
>
> For the relatively short TempCompass benchmark, selecting query-relevant and diversified keyframes results in considerable overlap with uniform sampling, leading to limited performance improvement. In contrast, on the more challenging Video_Holmes benchmark, our approach of carefully selecting keyframes and incorporating threaded narratives significantly enhances the MLLM's video reasoning capabilities.

---

> > ### Author Response · Authors · 2025-11-23
> >
> > ### W2: [**Results with more recent VLMs**]
> > As suggested by the reviewer, we conduct further evaluations on more recent VLMs including **Qwen2.5-VL-7B**, **Qwen3-VL**, and **InternVL3-8B**. Notably, the Qwen3-VL-8B model was released on November 1st. Detailed results are presented below.
> > |Model|Short|Medium|Long|VideoMME (no sub / sub)|LVB|MLVU|
> > |---|:---:|:---:|:---:|---|:---:|:---:|
> > |Qwen2.5-VL-7B|65.9 / 66.4|54.4 / 54.3|45.8 / 46.9|55.4 / 55.9|52.7|55.8|
> > |**+ KFC**|68.8 / 70.7|52.6 / 54.9|49.3 / 51.4|**56.9 / 59.0**|**54.3**|**62.6**|
> > |**+ Nar-KFC**|70.1 / 71.0|54.4 / 55.2|49.0 / 49.4|**57.9** / *58.6*|**55.3**|**64.4**|
> > ||||||||
> > |InternVL3-8B|68.7 / 70.9|58.3 / 58.2|50.0 / 50.9|59.0 / 60.0|53.6|60.9|
> > |**+ KFC**|70.9 / 71.9|60.6 / 60.1|50.9 / 51.8|**60.8 / 61.4**|**54.5**|**67.5**|
> > |**+ Nar-KFC**|72.9 / 73.9|62.9 / 62.7|55.7 / 55.8|**63.8 / 64.1**|**54.8**|**68.4**|
> > ||||||||
> > |Qwen3-VL-8B|68.4 / 70.7|55.4 / 55.3|50.1 / 52.0|58.0 / 59.1|54.7|49.5*|
> > |**+ KFC**|68.4/71.9|57.3/57.0|50.8/50.7|**58.9 / 59.9**|**55.8**|**63.0**|
> > |**+ Nar-KFC**|70.4 / 72.9|60.1 / 59.7|52.7 / 52.4|**61.1 / 61.7**|**56.2**|**65.8**|
> >
> > We limit the maximum frame resolution to 256$\times$28$\times$28 during evaluation on the Qwen3-VL-8B model to avoid generating an excessively large number of tokens (e.g., >16,384). **\*** : For the Qwen3-VL baseline result on the MLVU benchmark (i.e., 49.5), we found that the model often refuses to select an option from the given candidates, outputting “None of the options can answer the question since the provided video frames are not sufficient.” This leads to many incorrect answers and significantly lower performance. In contrast, our KFC-selected keyframes effectively address this issue by providing query-relevant video content.
> >
> > ***
> > ### W3: [**Baseline results verification**]
> > We believe there may be a misunderstanding regarding our reported results. As stated in Line 325, all performances reported in Table 1 are based on 8 uniformly sampled frames. This evaluation protocol is consistent with previous literature, where **8-frame inference** is commonly used. Our results are generally very close to, and aligned with, those reported in related works, even though the evaluation frameworks may differ (e.g., VLMEvalKit, LMMs-Eval).
> > |Model(baseline results on 8 frames)|Source|Venue|VideoMME(no sub/sub)|LVB|MLVU|
> > |---|---|:---|:---|---:|---:|
> > |LLaVA-OneVision-7B|Frame-Voyager [1]|ICLR2025|53.3 / -|-|58.5|
> > ||BOLT [2]|CVPR2025|53.8 / -|54.2|58.9|
> > ||MDP3 [3]|ICCV2025|53.6 / 53.9|54.2|59.3|
> > ||FrameOracle [4]|arXiv2025|53.8 / -|54.3|58.4|
> > ||**Ours reported**|-|53.3 / 55.9|54.5|58.5|
> > ||||||||
> > |LLaVA-Video-7B|BOLT [2]|CVPR2025|56.0 / -|-|-|
> > ||FrameOracle [4]|arXiv2025|55.9 / -|54.2|60.5|
> > ||**Ours reported**|-|55.9 / 56.7|54.2|60.5|
> > ||||||||
> > |InternVL2-8B|BOLT [2]|CVPR2025|52.6/-|-|-|
> > ||**Ours reported**|-|51.9/52.5|52.3|54.3|
> >
> > Additionally, we report quantitative baseline results with 64-frame inference, as we also include experiments with varying numbers of frames in Fig. 3 of the main paper. We collect baseline model results across three benchmarks and find that our re-implemented results are also consistent with recent literature.
> > |Model(baseline results on 64 frames)|Source|Venue|VideoMME(no sub/sub)|LVB|MLVU|
> > |---|---|:---:|---|:---:|:---:|
> > |LLaVA-Video-7B|AKS [5]|CVPR2025|64.4 / -|58.9|-|
> > ||FlowAgent [6]|ICCV2025|62.6 / -|58.2|70.8|
> > ||TrialTriumph [7]|ICCV2025|63.3 / -|58.2|70.8|
> > ||**Ours reported**|-|63.6 / 67.7|58.8|70.8|
> >
> > [1] Frame-Voyager: Learning to Query Frames for Video Large Language Models. ICLR 2025.
> > [2] BOLT: Boost Large Vision-Language Model Without Training for Long-form Video Understanding. CVPR 2025.
> > [3] MDP3: A Training-free Approach for List-wise Frame Selection in Video-LLMs. ICCV 2025.
> > [4] FrameOracle: Learning What to See and How Much to See in Videos. arXiv 2025.
> > [5] Adaptive Keyframe Sampling for Long Video Understanding. CVPR 2025.
> > [6] Flow4Agent: Long-form Video Understanding via Motion Prior from Optical Flow. ICCV 2025.
> > [7] From Trial to Triumph: Advancing Long Video Understanding via Visual Context Sample Scaling and Self-reward Alignment. ICCV 2025.

---

> > > ### Comment · Reviewer_A1NG · 2025-11-24
> > > **Rebuttal Response**
> > >
> > > Thank you for the authors’ detailed response. The rebuttal has addressed most of my concerns. However, regarding the 8-frame inference, I still find it difficult to agree with the authors’ justification:
> > >
> > > (1) For long-video scenarios, can 8 frames truly cover all relevant content, especially for summarization-type questions?
> > >
> > > (2) As modern models support increasingly long context windows—and fps-based sampling has already become mainstream—is it still meaningful to maintain an 8-frame baseline?
> > >
> > > (3) If the argument is based on efficiency, note that dense captioning and frame selection themselves introduce non-trivial overhead. Moreover, LLaVA-Video directly inputs 64 frames, resulting in 13×13×64 = 10,816 visual tokens, which is roughly comparable to the efficiency shown in Table 5.
> > >
> > > That said, I still feel that the overall quality of the paper is above the acceptance threshold, and therefore I am maintaining my score.

---

> > > > ### Author Response · Authors · 2025-11-26
> > > >
> > > > We sincerely appreciate the reviewer’s valuable feedback and fully respect your judgement. For the sake of constructive discussion (not for other purposes), we would like to further address the three newly raised points below.
> > > >
> > > > ***
> > > > (1) We completely agree with the reviewer that using only 8 frames is insufficient to cover all relevant content in long videos. In fact, on the MLVU benchmark—where our method achieves the most significant improvements—we observed that the **topic** task (which corresponds to the **summarization** task mentioned by the reviewer) is the only one where KFC resulted in a decrease in performance.
> > > > |MLVU task type|Uniform|KFC|Nar-KFC|
> > > > |---|:---:|:---:|:---:|
> > > > |plotQA|56.8|58.8|66.2|
> > > > |needle|50.7|78.3|78.9|
> > > > |ego|50.9|55.4|55.0|
> > > > |count|35.9|46.1|46.7|
> > > > |order|42.1|52.5|53.3|
> > > > |anomaly|64.0|65.0|66.0|
> > > > |topic|77.9|**76.0**|83.3|
> > > >
> > > > While our Nar-KFC significantly improves performance by providing more comprehensive coverage of video content.
> > > >
> > > > (2) We maintain the 8-frame setting in Table 1 to ensure a relatively fair comparison with many previous methods, as 8 frames is a commonly reported setting in the literature. Additionally, we scale the number of frames and use larger models (72B) in Figure 3 and Tables 8 and 9. We believe that achieving comparable or even superior performance with fewer frames remains a challenging and meaningful topic for future research.
> > > >
> > > > (3) Regarding efficiency, we find the calculation of 10,816 visual tokens by the reviewer somewhat confusing. Does the "13×13" mentioned refer to the patch resolution, rather than the number of visual tokens? Converting a frame into only 169 tokens in LLaVA-Video does not seem accurate. According to the FrameOracle paper (https://arxiv.org/pdf/2510.03584), feeding 16 frames into LLaVA-Video-7B results in 11,644 visual tokens (in our paper, we report 6,280 visual tokens with 8 frames in InternVL2), and 64 frames should correspond to 11,644$\times$4=**46,576** tokens, which is significantly larger than the 210 narratives (4,725) introduced in our method.

---

> ### Author Response · Authors · 2025-11-23
>
> ### W4: [**Discussion with recent two long video works**]
> We thank the reviewer for highlighting these two recent and related works. Since ICCV 2025 is scheduled in October and the Flow4Agent paper was released on October 7th—both after the ICLR submission deadline (September 24th). That’s why we missed these two related works.
>
> Flow4Agent introduces a creative approach by leveraging optical flow to extract motion priors and filter out irrelevant information in long videos. TrialTriumph improves long video inference by self-rewarding various frame combinations within a rich visual context, thereby selecting the best video predictions. A comparative summary with these works is presented below:
> |Model|Frame|VideoMME(no sub)|LVB|MLVU|
> |---|:---:|:---:|:---:|:---:|
> |LLaVA-Video-7B|64|62.6|58.2|70.8|
> |w/ **Flow4Agent**|64|64.7|60.4|71.4|
> |w/ **TrialTriumph**|32|**66.5**|**61.4**|**73.4**|
> |w/ **Nar-KFC (ours)**|32|65.1|61.1|73.3|
>
>
> Our Nar-KFC generally outperforms Flow4Agent, but does not reach the performance of TrialTriumph. We believe this is mainly due to TrialTriumph’s mechanism of self-rewarding the best frame combinations from multiple inference answers.
>
> We are revising our next version paper and will ensure to cite and discuss these very recent works.

---

### Author Response · Authors · 2025-12-01
**Revised Manuscript Uploaded and General Response**

We sincerely thank all reviewers for their constructive feedback and thoughtful suggestions. We greatly appreciate the recognition of our work’s *novelty* (TdJ3), *significance* (A1NG, TdJ3, co9R, N1Wd), and *quality* (A1NG, co9R, N1Wd).


To address the major concerns raised by the reviewers, primarily regarding **verification with newer MLLMs** and **additional benchmarks**, we have thoroughly revised the manuscript. All updated sections are clearly marked in **blue** for your convenience. The main revisions include:

- **Integration of newer MLLMs**, such as InternVL3-8B, Qwen2.5-VL-7B (Tab. 1), and Qwen3-VL-8B (App. Tab. 8).

- **Expanded results on open-ended and fine-grained generation tasks**, including MMBench-Video and MLVU-OpenEnded (Tab. 2 & Lines 368-374), as well as  **additional QA results** on TempCompass and Video-Holmes (App. Tab. 12 & Lines 991-998).

In addition, we have addressed specific concerns raised by individual reviewers:

- **Verified baseline results** and **incorporated two ICCV 2025 works** as suggested by Reviewer A1NG.

- **Quantifying failure analysis** in response to Reviewer TdJ3.

- Included a discussion of **computational overhead** (App. Sec D.5) and **comparisons with token-reduction-based VideoLLMs** (App. Sec D.2), as suggested by Reviewers co9R and N1Wd. (We would like to clarify that these results were already presented in our initial submission.)

We believe these revisions have significantly strengthened the paper, and we hope our responses and the additional experimental results fully address the concerns.


Sincerely,
Authors of Paper 16552

---

### Meta-Review · Area_Chair_ytg6 · 2026-01-12

**Summary:**

Reviewers’ concerns were primarily about experimental completeness and practicality rather than fundamental soundness. Two reviewers initially scored the paper at 4 due to (i) limited task/benchmark diversity (QA-only; need captioning/open-ended tasks), (ii) use of outdated backbones and lack of validation on newer MLLMs, and (iii) questions about the cost/overhead and possible noise/hallucination introduced by query-agnostic narratives. Another reviewer at 6 raised concerns about limited benchmarks, baseline verification, and missing discussion of very recent related works, while also questioning whether an 8-frame evaluation setting is still meaningful for long-video scenarios.

The rebuttal addressed these points with substantial additions: evaluations on newer MLLMs (e.g., Qwen2.5/3-VL, InternVL3), expanded tasks/benchmarks including captioning and open-ended generation (e.g., MMBench-Video, MLVU-OpenEnded, Dream1K), added token-reduction method comparisons and larger-model scaling results, and provided overhead analysis plus failure-style breakdowns suggesting limited negative impact from narratives. With one reviewer explicitly raising from 4→6 after the added captioning evidence and others maintaining 6 while stating most concerns were addressed, as Area Chair I recommend acceptance (lean accept).

**Reviewer Concerns:**

Reviewer N1Wd — 4 → 6

Addressed: task diversity concern via added captioning (Dream1K) and additional non-QA evaluations; validation on newer MLLMs; added scaling evidence.
Outstanding: mainly whether all new results are fully integrated/clearly presented in the final version (not a technical blocker).

Reviewer A1NG — 6 (maintain)

Addressed: broader benchmarks, newer MLLMs, baseline result verification, and discussion of recent related works.
Outstanding: skepticism about the continued relevance of 8-frame baselines for long-video settings and possible under-accounting of preprocessing overhead (still a minor concern, but reviewer maintained 6).

Reviewer co9R — 6 (no follow-up)

Addressed (in rebuttal): concerns about missing subtle/local changes (added fine-grained captioning evidence), captioner cost (overhead profiling), and lack of comparison to token-reduction methods (added).
Outstanding: efficiency tradeoff of narration in truly on-demand settings remains somewhat assumption-dependent; no explicit reviewer confirmation.

Reviewer TdJ3 — 4 (no follow-up)

Addressed (in rebuttal): quantified “noise/hallucination” risk via task-type breakdown; tested alternative selectors beyond CLIP and evaluated on newer MLLMs.
Outstanding: inherent risk of query-agnostic narratives injecting irrelevant/incorrect text is mitigated but not eliminated; still a conceptual weakness without reviewer confirmation.

**Reviewer Scores:**

See the box above.

---

### Decision · Program_Chairs · 2026-01-26

Accept (Poster)